# Three-Dimensional Trajectory Prediction with 3DMoTraj Dataset

**Hao Zhou** [* 1 2 3]  **Xu Yang** [* 4]  **Mingyu Fan** [5]  **Lu Qi** [6]  **Xiangtai Li** [7]  **Ming-Hsuan Yang** [8]  **Fei Luo** [1 3]

## Abstract

With the growing interest in embodied and spatial intelligence, accurately predicting trajectories in 3D environments has become increasingly critical. However, no datasets have been explicitly designed to study 3D trajectory prediction. To this end, we contribute a 3D motion trajectory (3DMoTraj) dataset collected from unmanned underwater vehicles (UUVs) operating in oceanic environments. Mathematically, trajectory prediction becomes significantly more complex when transitioning from 2D to 3D. To tackle this challenge, we analyze the prediction complexity of 3D trajectories and propose a new method consisting of two key components: decoupled trajectory prediction and correlated trajectory refinement. The former decouples inter-axis correlations, thereby reducing prediction complexity and generating coarse predictions. The latter refines the coarse predictions by modeling their inter-axis correlations. Extensive experiments show that our method significantly improves 3D trajectory prediction accuracy and outperforms state-of-the-art methods. Both the 3DMoTraj dataset and the method are available at https://github.com/zhouhao94/3DMoTraj.

## 1. Introduction

In recent years, the embodied (Wang et al., 2024; Kaur et al., 2023; Qi et al., 2024) and spatial intelligence (Huang et al., 2024) have witnessed rapid advancement, both of

---
[*]Equal contribution  [1]School of Computing and Information Technology, Great Bay Institute for Advanced Study/Great Bay University, Dongguan, China.  [2]Tsinghua Shenzhen International Graduate School, Tsinghua University, Shenzhen, China. [3]Dongguan Key Laboratory for Intelligence and Information Technology, Dongguan, China. [4]State Key Laboratory of Multimodal Artificial Intelligence Systems, Institute of Automation, Chinese Academy of Sciences, Beijing, China. [5]Institute of Artificial Intelligence, Donghua University, Shanghai, China. [6]Insta360 Research, Shenzhen, China. [7]Nanyang Technological University, Singapore. [8]University of California Merced, Merced, USA. Correspondence to: Fei Luo <luofei@gbu.edu.cn>.

*Proceedings of the 42nd International Conference on Machine Learning*, Vancouver, Canada. PMLR 267, 2025. Copyright 2025 by the author(s).

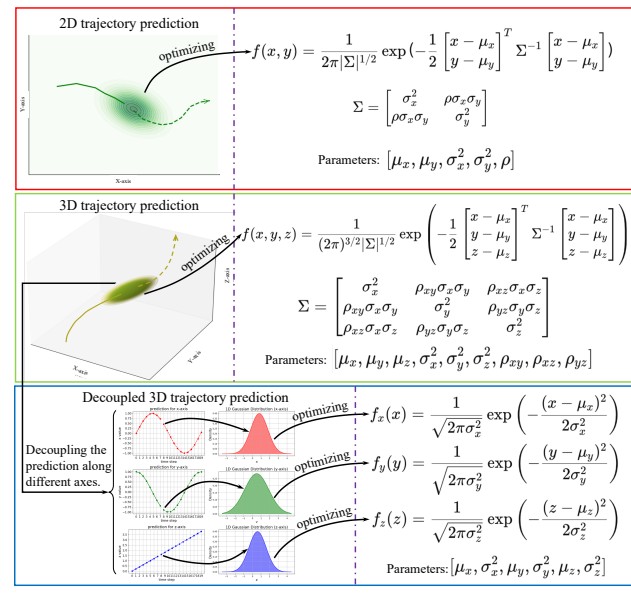

Figure 1: Illustrations of different trajectory predictions and their prediction complexities. The 2D, 3D, and decoupled 3D predictions for each point correspond to optimizing one 2D, one 3D, and three 1D Gaussian distributions, which involve optimizing 5, 9, and 6 free parameters, respectively.

which emphasize the ability to understand, reason about, and manipulate information within a three-dimensional (3D) space (Zhu et al., 2025; Ishida, 2024). Such advancement highlights the critical role of 3D trajectory prediction, which enables systems to effectively navigate and interact with dynamic environments by anticipating future movements in space. For example, in unmanned systems, accurate 3D motion path prediction for unmanned underwater vehicles (UUVs) (Cao et al., 2022) and unmanned aerial vehicles (UAVs) (Zhang et al., 2023; Huang et al., 2023) helps them avoid obstacles and plan efficient navigation routes in ocean and air environments.

Despite the significant importance of 3D trajectory prediction for embodied and spatial intelligence, most current trajectory prediction tasks, including pedestrian (Mohamed et al., 2020; Mangalam et al., 2020; Zhou et al., 2021) and vehicle (Gao et al., 2020; Zhou et al., 2023; Shi et al., 2022) trajectory prediction, focus primarily on 2D trajectories due to the scarcity of publicly available 3D trajectory datasets. To this end, we contribute a 3D motion trajectory (3DMo-

Traj) dataset, which consists of 3D trajectories collected from UUVs across eight distinct oceanic scenarios. There are two reasons for collecting trajectories from UUVs: (1) the trajectories of multiple UUVs exhibit complex interactions arising from formation movements, and (2) ocean currents introduce fluctuations in UUV trajectories, significantly increasing the challenge of accurate prediction.

The 3DMoTraj dataset is annotated with frame-wise intentions to facilitate future research. Existing 2D trajectory prediction methods have shown that incorporation of intentions (Rasouli et al., 2019; Liu et al., 2020; Mangalam et al., 2021; Choi et al., 2021; Zhao et al., 2021) can significantly improve prediction accuracy. Specifically, the trajectories in 3DMoTraj are annotated with both static and motion intentions. The static intention is defined as the octant the agent is expected to reach by the end of the prediction horizon, whereas the motion intention represents the mean velocity changes between the observed and predicted trajectories. Static intentions provide insight into the agent's final spatial goal, while motion intentions capture the agent's dynamic state transitions.

Aside from the scarcity of available datasets, one major challenge impeding progress in 3D trajectory prediction is its substantially higher prediction complexity. As illustrated in Figure 1, predicting a 2D trajectory is generally equivalent to optimizing 2D Gaussian distributions with 5 parameters to approximate the possible locations of each 2D point (Mohamed et al., 2020). In contrast, 3D trajectory prediction requires optimizing a 3D Gaussian distribution with 9 parameters for each 3D point. As a result, the complexity of predicting one 3D point is approximately twice that of one 2D point. Furthermore, the prediction complexity of a 3D scenario involving $N$ points is linearly related to $9 \times N$, significantly hindering the prediction of complex scenarios containing multiple 3D agents. Therefore, reducing the optimized parameters for each 3D point is important.

To address the increased prediction complexity of 3D trajectory, we propose a novel 3D trajectory prediction method comprising two key components: decoupled trajectory prediction and correlated trajectory refinement. Optimizing 3D Gaussian distribution can be decomposed into optimizing three individual 1D Gaussian distributions and modeling their correlations; please refer to Section A in the Appendix for detailed mathematical proofs. We adopt a divide-and-conquer strategy. Specifically, 3D trajectory prediction is first decoupled along three independent axes to mitigate the impact of inter-axis correlations, followed by modeling these correlations. In this way, the prediction of each 3D point with 9 parameters is first simplified as the optimization of three 1D Gaussian distributions with 6 free parameters, as illustrated in Figure 1, thereby reducing the overall prediction complexity. Therefore, the decoupled trajectory pre-

diction part employs three independent decoders to predict future trajectories along the x-, y-, and z-axes, producing a coarse prediction. The correlated trajectory refinement part then models the inter-axis correlations of the coarse predictions and generates offsets to refine them. Extensive experiments conducted on the 3DMoTraj dataset demonstrate the superior performance of our proposed method for 3D trajectory prediction.

In conclusion, the contributions of this work are as follows:

- We propose a 3D trajectory dataset named 3DMoTraj, which is collected from UUVs in ocean environments. The dataset includes frame-wise annotations for both motion and static intentions.
- We analyze the increased prediction complexity of 3D trajectories compared to 2D trajectories and propose a 3D trajectory prediction method that consists of two components: decoupled trajectory prediction and correlated trajectory refinement. The former decouples the inter-axis correlations to generate coarse predictions. The latter models the inter-axis correlations to refine these coarse predictions.
- We conduct extensive experiments on the 3DMoTraj dataset. The experimental results verify the superior performance of our method in 3D trajectory prediction, establishing a solid baseline for future research.

## 2. Related Work

**Trajectory Prediction Datasets.** Some trajectory datasets are collected from top-view cameras, such as ETH/UCY (Lerner et al., 2007; Pellegrini et al., 2009), PWRD, and NYGC (Zhou et al., 2011), which focus on pedestrian trajectory prediction. Other datasets, such as SDD (Robicquet et al., 2016) and inD (Bock et al., 2020), are collected using flying drones and include trajectories of vehicles, bicycles, and pedestrians. With advances in sensor technology, high-quality datasets such as Apollo (Ma et al., 2019), Argoverse (Chang et al., 2019; Wilson et al.), and Waymo (Ettinger et al., 2021) use sensors, such as LiDAR, radar, high-definition maps, and localization modules, to record trajectories. Recognizing the importance of agents' intentions in prediction, datasets like PIE (Rasouli et al., 2019) and LOKI (Girase et al., 2021) include frame-wise intention annotations. PIE focuses on short-term pedestrian intentions, while LOKI provides long-term intentions for hybrid road users. Although these datasets offer valuable resources for trajectory prediction, they mainly focus on 2D scenarios. To date, no publicly available datasets are specifically designed for 3D trajectory prediction.

**Trajectory Prediction Methods.** These methods in 2D scenarios can be broadly categorized into RNN-based methods, one-shot prediction methods, and intention-conditioned methods. RNN-based methods iteratively predict future trajectory points. Representative works include S-GAN (Gupta

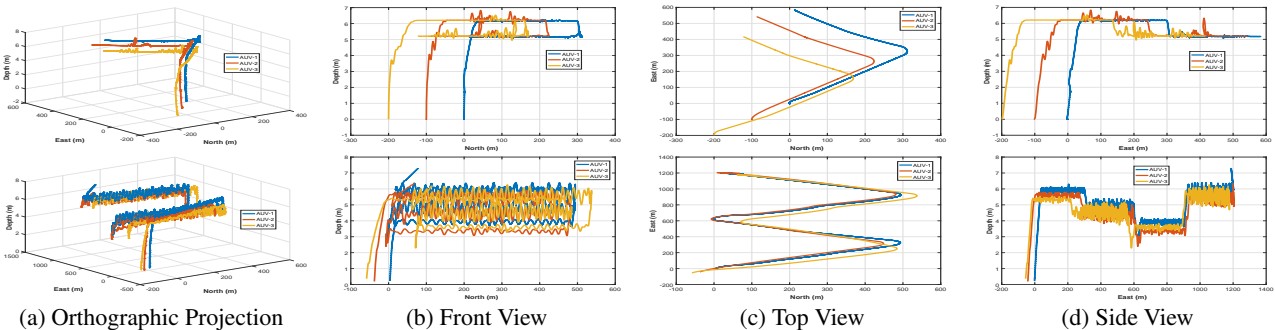

| (a) Orthographic Projection | (b) Front View | (c) Top View | (d) Side View |

Figure 2: Orthographic projections, along with front, top, and side views of 3D trajectories for two typical underwater scenarios. Please refer to Figure 9 in the Appendix for the rest of the scenarios.

et al., 2018), PIE (Rasouli et al., 2019), STGAT (Huang et al., 2019), Trajectron++(Salzmann et al., 2020), and SR-LSTM(Zhang et al., 2020). One-shot prediction methods predict the entire future trajectory in a single step. Prominent approaches include SSTGCNN (Mohamed et al., 2020), MSRL (Wu et al., 2023), MRGTraj (Peng et al., 2023), S-Implicit (Mohamed et al., 2022), and FlowChain (Maeda & Ukita, 2023). Intention-conditioned methods predict future intentions and then generate trajectories conditioned on these predictions. Notable examples include LBEBM (Pang et al., 2021), TNT (Zhao et al., 2021), NSP-SFM (Yue et al., 2022), and MemoNet (Xu et al., 2022). Besides, interaction modeling techniques such as pooling (Mangalam et al., 2020), attention (Salzmann et al., 2020), and graph neural networks (Mohamed et al., 2020) are studied to capture the dynamics of multi-agent systems. Recent research also addresses real-world challenges, including domain adaptation (Feng et al., 2024) and observation length shift (Xu & Fu, 2024), which commonly arise in practical deployment. Furthermore, large language models and unsupervised learning applied in time-series tasks (Sun et al., 2024; Yang & Hong, 2022) also inspire research in trajectory prediction.

Trajectory prediction in 3D scenarios primarily focuses on predicting UAV trajectories. Current research on UAV trajectory prediction largely relies on traditional methods, such as Gaussian process regression (Xie & Chen, 2022), Kalman filters (Wang et al., 2018), and principal component analysis (Zhang et al., 2023), as well as simple learning-based methods, including LSTMs (Zhong et al., 2022) and CNNs (Liu et al., 2021). However, these methods struggle to predict 3D trajectories in complex scenarios due to the high prediction complexity of 3D trajectories.

## 3. 3D Motion Trajectory Dataset

The 3DMoTraj dataset comprises eight scenarios featuring three-dimensional (3D) trajectories collected from unmanned underwater vehicles (UUVs) operating in ocean environments. The 3D trajectories are recorded using positioning devices installed on the UUVs. Each scenario includes three UUVs following predefined formations. We

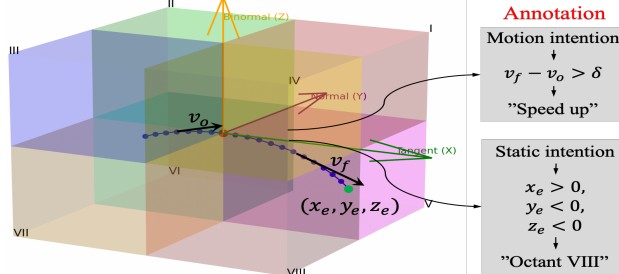

Figure 3: Illustration of annotations for motion and static intentions. Motion intention is determined by the velocities of the observed and predicted trajectories, while static intention is based on the location of the trajectory's endpoint.

present the orthographic projections and the front, top, and side views of the 3D trajectories for two representative scenarios in Figure 2. Notably, in addition to the complex 3D motion patterns of each UUV, the trajectories also exhibit fluctuations caused by ocean currents, which introduce significant challenges to the prediction task. This section provides an overview of the 3DMoTraj dataset and analyzes its statistical characteristics.

### 3.1. Dataset Construction

To enhance the diversity of 3D motion behaviors, we collect trajectory data using three agents over an extended period.

**Dataset Preprocess.** The sampling rate for the trajectory points is 10 fps, resulting in a very short time interval between consecutive points and, thus, minimal spatial separation. To address this issue, we first downsample the trajectory points to a sample rate of 2 fps. Since each scenario includes only three agents, we further divide the trajectory data into short fragments, each containing 200 data frames. We assign unique IDs to agents in each fragment to enhance the representation of agents in the dataset. The fragments for each scenario are then randomly divided into training, validation, and test sets with a 1:1:1 ratio.

**Dataset Annotation.** Intentions play a crucial role in trajectory prediction. We annotate intentions in the dataset to support further research in 3D trajectory prediction.

| Scenario | #1 | #2 | #3 | #4 | #5 | #6 | #7 | #8 |
|---|---|---|---|---|---|---|---|---|
| Frame number | 765 | 1744 | 1010 | 1558 | 1861 | 895 | 3379 | 2597 |
| Distance(m) | 0.707±0.282 | 0.482±0.212 | 0.505±0.185 | 0.569±0.287 | 0.491±0.202 | 0.784±0.333 | 0.707±0.251 | 0.514±0.273 |
| Velocity(m/s) | 1.413±0.564 | 0.964±0.424 | 1.009±0.369 | 1.139±0.573 | 0.982±0.405 | 1.567±0.666 | 1.415±0.501 | 1.028±0.546 |
| Acceleration(m/s²) | 0.003±1.373 | 0.002±1.046 | 0.005±0.937 | 0.002±1.361 | 0.003±1.045 | -0.001±1.683 | 0.001±1.433 | 0.001±1.212 |
| Curvature(1/m) | 0.184±2.428 | 0.419±3.494 | 0.215±2.784 | 0.486±3.591 | 0.185±2.692 | 0.074±0.518 | 0.030±0.300 | 0.407±2.259 |
| Motion intention (velocity change) | ↑:679,↓:1028, →:440 | ↑:1494,↓:1955, →:1450 | ↑:997,↓:1002, →:816 | ↑:1735,↓:1350, →:1293 | ↑:1559,↓:1981, →:1673 | ↑:948,↓:1182, →:370 | ↑:3168,↓:4810, →:1530 | ↑:2562,↓:3045, →:1703 |
| Static intention (Endpoint octant) | #1:390,#2:104, #3:722,#4:5, #5:288,#6:138, #7:445,#8:55 | #1:161,#2:16, #3:144,#4:95, #5:1051,#6:850, #7:2141,#8:441 | #1:74,#2:509, #3:669,#4:491, #5:11,#6:398, #7:312,#8:351 | #1:175,#2:1333, #3:509,#4:975, #5:57,#6:713, #7:196,#8:378 | #1:301,#2:902, #3:826,#4:854, #5:300,#6:422, #7:911,#8:697 | #1:324,#2:524, #3:291,#4:657, #5:145,#6:218, #7:47,#8:294 | #1:788,#2:594, #3:682,#4:951, #5:1772,#6:878, #7:2579,#8:1264 | #1:1363,#2:1243, #3:2182,#4:786, #5:412,#6:474, #7:556,#8:294 |

Table 1: Statistics information of the 3DMoTraj dataset. For motion intention, notations ↑, ↓, and → respectively represent speed up, speed down, and constant speed. For static intention, notation $#N$ denotes the endpoints in $N$-th octant.

Unlike datasets such as PIE (Rasouli et al., 2019) and LOKI (Girase et al., 2021), which annotate intentions based solely on future trajectories, we propose utilizing both observed and future trajectories. Specifically, we compare the average velocity of the observed trajectory ($v_o$) and the future trajectory ($v_f$) to label samples with motion intention 'speed up,' 'slow down', or 'constant speed', as shown in Figure 3. A trajectory sample is labeled as 'speed up' if $v_f - v_o > \delta$, 'speed down' if $v_f - v_o < -\delta$, and 'constant speed' if $|v_f - v_o| \leq \delta$, with $\delta$ set to 0.1 by default.

Additionally, we annotate future location intention, referred to as static intention, based on the endpoint of each trajectory sample, as illustrated in Figure 3. Specifically, we define the current point as the origin, with the velocity direction at that point serving as the x-axis. Using the Frenet-Serret frame, we establish a local coordinate system, within which the endpoints are classified into one of eight octants. This octant-based labeling provides trajectory samples with static intention. As a result, our dataset includes annotations for both motion and static intentions.

**Dataset Usage.** Following the setting in datasets ETH (Pellegrini et al., 2009) and UCY (Lerner et al., 2007), we adopt a sliding window technique to generate samples from the dataset. Each sample contains 20 frames of trajectory points, with the first 8 frames (4 seconds) as the observed trajectory and the remaining 12 frames (6 seconds) as the ground truth future trajectory. Furthermore, we recommend evaluating prediction methods using a leave-one-out cross-validation strategy. Specifically, the model is trained and validated on the training and validation sets of seven scenarios and tested on the test set of the remaining scenario. This strategy helps mitigate validation bias.

### 3.2. Dataset Statistics

To demonstrate the practicality of the proposed dataset for evaluating 3D trajectory prediction, we analyze its characteristics across several dimensions. A detailed summary of the dataset's statistical properties is presented in Table 1.

**Basic information.** Firstly, we present the number of frames for each processed scenario in the second row. Our dataset

contains over 13000 frames, offering sufficient data for model training and validation.

**Motion information.** Secondly, we present the average distance, velocity, and acceleration of the trajectory points in the third to fifth rows. The average distance across all scenarios ranges from 0.4 to 0.8 meters, with corresponding average velocities between 0.9 and 1.6 meters per second, which are relatively high speeds for UUVs in ocean environments. Additionally, the significant standard deviations in velocity and acceleration indicate frequent velocity changes among agents in our dataset. The distributions of distance, velocity, and acceleration, presented in the Appendix, further illustrate this variability. These characteristics make the dataset well-suited for evaluating the ability of prediction methods to handle complex velocity dynamics.

**Curvature information.** Thirdly, we calculate the average 3D curvature of the trajectory points, reported in the sixth row. The lowest average curvature is 0.074 in scenario 1, while the highest is 0.486 in scenario 4. Notably, the standard deviation of curvature in each scenario is approximately ten times its mean value. The curvature distributions across all scenarios, visualized in the Appendix, further support this variability. These statistics indicate that the trajectories in our dataset exhibit a wide range of curvatures, providing a robust benchmark for evaluating the prediction ability of methods on complex curvilinear paths.

**Intention information.** Finally, the statistics on intention information are presented in the seventh and eighth rows. Motion intentions are evenly distributed across the labels 'speed up,' 'slow down,' and 'constant speed.' Similarly, static intentions are uniformly distributed across eight different octants. These balanced distributions ensure that our dataset offers a clear distinction in intention labeling.

## 4. 3D Trajectory Prediction Method

In this section, we address 3D trajectory prediction in two steps: *Decoupling inter-axis correlations* by independently predicting the trajectory along each axis, and *Modeling inter-axis correlations* to generate offsets that refine the initial predictions. The former reduces the high prediction

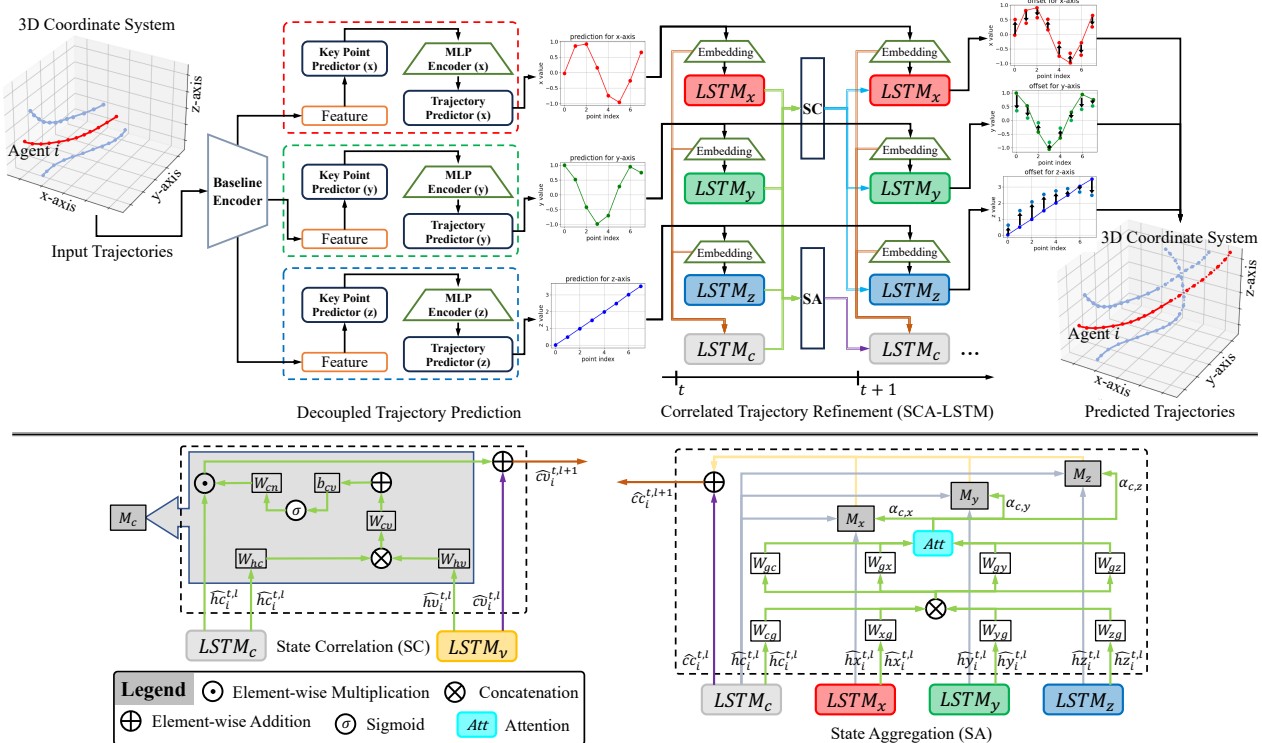

Figure 4: The framework of our proposed 3D trajectory prediction network. In addition to the baseline encoder from LBEBM, our method consists of two key components: decoupled trajectory prediction and correlated trajectory refinement. The former independently predicts the future trajectories along each axis, thereby decoupling inter-axis correlations. The latter generates offsets to refine these predictions by modeling their inter-axis correlations.

complexity of 3D trajectories, providing a coarse prediction. The latter refines these predictions by capturing their inter-axis correlations.

**Problem Formulation.** Let $p_i^t = (x_i^t, y_i^t, z_i^t) \in \mathbb{R}^3$ represent the position of agent $i$ at time $t$. The observed trajectory of agent $i$ is defined as $\mathbf{p}_i^o = \{p_i^t \mid t = 1, ..., t_o\}$, and the ground truth future trajectory is $\mathbf{p}_i^f = \{p_i^t \mid t = t_o + 1, ..., t_o + t_f\}$, where $t_o$ and $t_f$ are the time horizons of the observed and future trajectories, respectively. The objective of 3D trajectory prediction is to use the observed trajectory $\mathbf{p}_i^o$ as input to predict a future trajectory $\hat{\mathbf{p}}_i^f = \{\hat{p}_i^t \mid t = t_o + 1, ..., t_o + t_p\}$ that closely approximates the ground truth trajectory $\mathbf{p}_i^f$.

### 4.1. Decoupled Trajectory Prediction

We adopt the LBEBM method (Pang et al., 2021) as our baseline. To decouple the inter-axis correlations of 3D trajectories, we replace LBEBM's decoder with three independent decoders to predict trajectories separately along the x-, y-, and z-axes. LBEBM is a key point-conditioned prediction method that first predicts future key points and then generates the future trajectories conditioned on those points. We follow the same overall procedure as LBEBM but decouple the prediction process into three parallel streams for each axis. The detailed process of our decoupled prediction is

illustrated in Figure 4.

Let $F_i^e$ represent the encoded feature of $\mathbf{p}_i^o$, extracted by LBEBM while accounting for interactions and other elements. We first employ three key point predictors to predict key points along different axes:

$$\hat{\nu}_i^k = \Phi_\nu(F_i^e), \quad \nu \in \{x, y, z\}, \tag{1}$$

where $\Phi_\nu(\cdot)$ and $\hat{\nu}_i^k$ denote the MLP key point predictor and the predicted key points for the $\nu$-axis, respectively. We then extract features from the predicted key points:

$$F_i^k = \kappa([\hat{\mathbf{x}}_i^k; \hat{\mathbf{y}}_i^k; \hat{\mathbf{z}}_i^k]), \tag{2}$$

where $\kappa(\cdot)$ is an MLP encoder, $[\cdot; \cdot; \cdot]$ denotes concatenation, and $F_i^k$ is the extracted feature. Finally, we concatenate $F_i^k$ with $F_i^e$ and feed this concatenated feature into three trajectory predictors to generate the future trajectory for each axis:

$$\hat{\nu}_i^f = \Psi_\nu([F_i^k; F_i^e]), \quad \nu \in \{x, y, z\}, \tag{3}$$

where $\Psi_\nu(\cdot)$ and $\hat{\nu}_i^f$ represent the MLP predictor and the predicted future trajectory for the $\nu$-axis, respectively.

### 4.2. Correlated Trajectory Refinement.

We design a novel LSTM-based method, State-Correlation and Aggregation LSTM (SCA-LSTM), to model the inter-axis correlations of the coarse predictions and refine them.

An overview of the SCA-LSTM is shown in Figure 4. SCA-LSTM employs a centralized architecture comprising a center LSTM, node LSTMs, a state correlation (SC) module, and a state aggregation (SA) module. The center LSTM extracts summarized trajectory features across different axes, while the node LSTM extracts trajectory features for each axis. The SC module captures the correlation between the center and node LSTMs, using it to update the node features. Meanwhile, the SA module aggregates the state features from the node LSTMs to update those of the center LSTM.

**Inputs Representation.** Instead of directly using raw trajectories, we extract their embeddings as inputs for SCA-LSTM.

For the node LSTMs, we first concatenate the predicted future trajectory with the observed trajectory for each axis. Then, we calculate and concatenate the trajectory, velocity, and acceleration of each axis. This concatenated information is passed through MLP encoder to extract embeddings:

$$F_i^\nu = \Omega_\nu([\nu_i^u; \mathbf{v}\nu_i^u; \mathbf{a}\nu_i^u]), \quad \nu \in \{x, y, z\}, \tag{4}$$

where $\nu_i^u$, $\mathbf{v}\nu_i^u$, and $\mathbf{a}\nu i^u$ represent the trajectory, velocity, and acceleration on the $\nu$-axis, respectively. $\Omega\nu$ is the MLP encoder for the $\nu$-axis. $F_i^\nu$ represents the extracted feature, which is the input to the node LSTM for the $\nu$-axis.

For the center LSTM, we apply max- and average-pooling operations to combine the input representations from all axes:

$$\begin{aligned} F_i^m &= MaxPooling(F_i^x, F_i^y, F_i^z), \\ F_i^a &= AvgPooling(F_i^x, F_i^y, F_i^z), \\ F_i^c &= [F_i^m; F_i^a], \end{aligned} \tag{5}$$

where $F_i^c$ is the extracted input feature for the center LSTM.

**The Center and Node LSTMs.** The center and node LSTMs use vanilla LSTMs to extract features from the input representations:

$$\begin{aligned} cc_i^t, hc_i^t &= LSTM_c(F_i^c[t], hc_{t-1}, cc_{t-1}; \Theta_c), \\ c\nu_i^t, h\nu_i^t &= LSTM_\nu(F_i^\nu[t], h\nu_{t-1}, c\nu_{t-1}; \Theta_\nu), \nu \in \{x,y,z\}, \end{aligned} \tag{6}$$

where $LSTM_c$, $\Theta_c$, $cc_i^t$, and $hc_i^t$ represent the center LSTM, its parameters, cell state, and hidden state, while $LSTM_\nu$, $\Theta_\nu$, $c\nu_i^t$, and $h\nu_i^t$ denote the node LSTM for the $\nu$-axis and its components. For more details on the vanilla LSTM, please refer to Section B in the Appendix. The main difference between the LSTMs used in our method and vanilla LSTMs is the inclusion of the SC and SA modules, which refine their cell and hidden states.

**The SC Module.** The SC module takes two inputs: the hidden states and cell states from the center and node LSTMs. The output is the refined cell states for the node LSTMs. Mathematically, the SC module is formulated as follows:

$$\hat{c}\nu_i^{t,l+1} = M_c(\hat{hc}_i^{t,l}, \hat{h\nu}_i^{t,l}) + \hat{c}\nu_i^{t,l} \quad \nu \in \{x, y, z\}, \tag{7}$$

where $\hat{c}\nu_i^{t,l+1}$ denotes the refined cell states for the $\nu$-axis after $l+1$ iterations, and $M_c$ represents the message correlation function.

We design the message correlation function with a gated selection mechanism to adaptively model correlations and refine the node LSTM's cell states. Specifically, we first extract the input features for the gate from hidden states:

$$\begin{aligned} F_i^{hc,l} &= W_{hc}\hat{hc}_i^{t,l}, \\ F_i^{h\nu,l} &= W_{h\nu}\hat{h\nu}_i^{t,l}, \quad \nu \in \{x, y, z\}, \end{aligned} \tag{8}$$

where $W_{hc}$ and $W_{h\nu}$ are linear transformations. Then, we construct a correlation gate using the features $F_i^{c,l}$ and $F_i^{\nu,l}$ such that:

$$g_i^{c,\nu} = \sigma(W_{c\nu}[F_i^{hc,l}; F_i^{h\nu,l}] + b_{c\nu}), \quad \nu \in \{x, y, z\}, \tag{9}$$

where $g_i^{c,\nu}$ is the gate vector for the $\nu$-axis, $W_{c\nu}$ and $b_{c\nu}$ are linear transformation and bias, and $\sigma$ refers to the Sigmoid function. Using the gate vector, $M_c$ is formulated as:

$$M_c(\hat{hc}_i^{t,l}, \hat{h\nu}_i^{t,l}) = W_{cn}g_i^{c,\nu} \odot \hat{hc}_i^{t,l}, \quad \nu \in \{x, y, z\}, \tag{10}$$

where $W_{cn}$ is a linear transformation.

**The SA Module.** The SA module adopts a similar strategy to the SC module but operates in reverse, aggregating the hidden states from node LSTMs to update the center LSTM. Mathematically, the SA module is formulated as:

$$\hat{cc}_i^{t,l+1} = \sum_{\nu \in \{x,y,z\}} M_\nu(\hat{hc}_i^{t,l}, \hat{h\nu}_i^{t,l}) + \hat{cc}_i^{t,l}, \tag{11}$$

where $\hat{cc}_i^{t,l+1}$ is the refined cell state of the center LSTM, and $M_\nu$ is the message aggregation function for the $\nu$-axis.

To adaptively aggregate the hidden states of the node LSTMs, we design a message aggregation function that incorporates axis-wise attention and an aggregation gate. The attention mechanism assigns appropriate weights to different axes, while the aggregation gate extracts key features for effective integration.

To build the attention and aggregation gate, we first extract a global feature $F_i^{g,l}$ from the hidden states of all LSTMs:

$$F_i^{g,l} = ([W_{gc}\hat{hc}_i^{t,l}; W_{gx}\hat{hx}_i^{t,l}; W_{gy}\hat{hy}_i^{t,l}; W_{gz}\hat{hz}_i^{t,l}]), \tag{12}$$

where $W_{gc}$, $W_{gx}$, $W_{gy}$, and $W_{gz}$ are linear transformations. We then project the global embedding into different LSTMs' projection spaces:

$$\begin{aligned} F_i^{gc,l} &= W_{gc}F_i^{g,l}, \\ F_i^{g\nu,l} &= W_{g\nu}F_i^{g,l}, \quad \nu \in \{x, y, z\}, \end{aligned} \tag{13}$$

where $F_i^{gc,l}$ is for the center LSTM, and $F_i^{g\nu,l}$ is for the node LSTM of the $\nu$-axis, with $W_{gc}$ and $W_{q\nu}$ being linear transformations. Using these projected features, the attention weight can be computed as follows:

$$\alpha_{c\nu} = \frac{exp(W_q F_i^{gcl}(W_k F_i^{g\nu,l})^T)}{\sum\limits_{\beta \in \{x,y,z\}} exp(W_q F_i^{gc,l}(W_k F_i^{g\beta,l})^T)}, \nu \in \{x,y,z\}, \tag{14}$$

| Scenario | #1 | #2 | #3 | #4 | #5 | #6 | #7 | #8 | Mean |
|---|---|---|---|---|---|---|---|---|---|
| SSTGCNN† (Mohamed et al., 2020) | 2.86/5.19 | 1.05/1.47 | 1.08/1.65 | 2.23/3.83 | 1.15/1.34 | 2.25/4.00 | 4.49/8.28 | 2.18/3.30 | 2.16/3.63 |
| MSRL (Wu et al., 2023) | 3.72/5.42 | 0.62/0.73 | 0.56/0.61 | 1.69/2.05 | 1.85/2.50 | 1.54/2.84 | 1.14/1.73 | 0.90/1.15 | 1.50/2.13 |
| FlowChain (Maeda & Ukita, 2023) | 1.44/3.20 | 0.61/0.99 | 0.62/1.02 | 0.90/1.76 | 0.54/0.84 | 1.31/2.52 | 1.18/2.53 | 0.93/1.84 | 0.94/1.84 |
| PECNet (Mangalam et al., 2020) | 0.79/1.05 | 0.70/1.29 | 0.74/1.26 | 1.29/2.46 | 0.41/0.58 | 1.16/1.65 | 0.92/1.53 | 1.22/2.21 | 0.90/1.50 |
| LBEBM (Pang et al., 2021) | 0.72/0.98 | 0.52/0.80 | 0.64/1.11 | 0.94/1.87 | 0.35/0.57 | 1.02/1.67 | 1.27/2.35 | 1.25/2.42 | 0.84/1.47 |
| NPSN* (Bae et al., 2022) | 0.75/0.83 | 0.69/1.21 | 0.71/0.96 | 0.83/1.28 | 0.34/0.40 | 0.97/1.20 | 0.71/0.93 | 1.03/1.50 | 0.75/1.04 |
| TrajCLIP (Yao et al., 2024) | 0.56/0.94 | 0.40/0.69 | 0.37/0.70 | 1.32/2.81 | 0.33/0.58 | 0.79/1.27 | 1.04/2.02 | 0.85/1.69 | 0.71/1.34 |
| CausalHTP (Chen et al., 2021) | 0.69/1.29 | 0.45/0.78 | 0.45/0.78 | 0.79/1.40 | 0.49/0.87 | 1.11/1.95 | 0.97/1.96 | 0.74/1.40 | 0.71/1.30 |
| MS-TIP (Nath et al., 2024) | 0.62/1.16 | 0.61/1.16 | 0.58/1.15 | 0.78/1.42 | 0.57/1.11 | 0.84/1.50 | 0.77/1.44 | 0.79/1.52 | 0.70/1.31 |
| MRGTraj (Peng et al., 2023) | 0.63/1.29 | 0.34/0.54 | 0.33/0.55 | 0.94/1.89 | 0.42/0.83 | 0.99/1.73 | 1.06/2.44 | 0.84/1.63 | 0.69/1.36 |
| S-Implicit (Mohamed et al., 2022) | 0.54/0.87 | 0.43/0.72 | 0.47/0.84 | 1.13/2.31 | 0.40/0.68 | 0.85/1.38 | 0.62/1.06 | 0.98/1.92 | 0.68/1.22 |
| Our | **0.36/0.51** | 0.37/0.60 | 0.48/0.86 | 0.81/1.69 | **0.28/0.44** | **0.69/1.10** | 1.12/1.95 | **0.55/0.99** | **0.58/1.02** |

Table 2: Comparison with state-of-the-art methods on the 3DMoTraj dataset under *best-of-20* evaluation setting. †: SSTGCNN uses predicted Gaussian distributions to sample future trajectories. *: NPSN here adopts PECNet as the baseline.

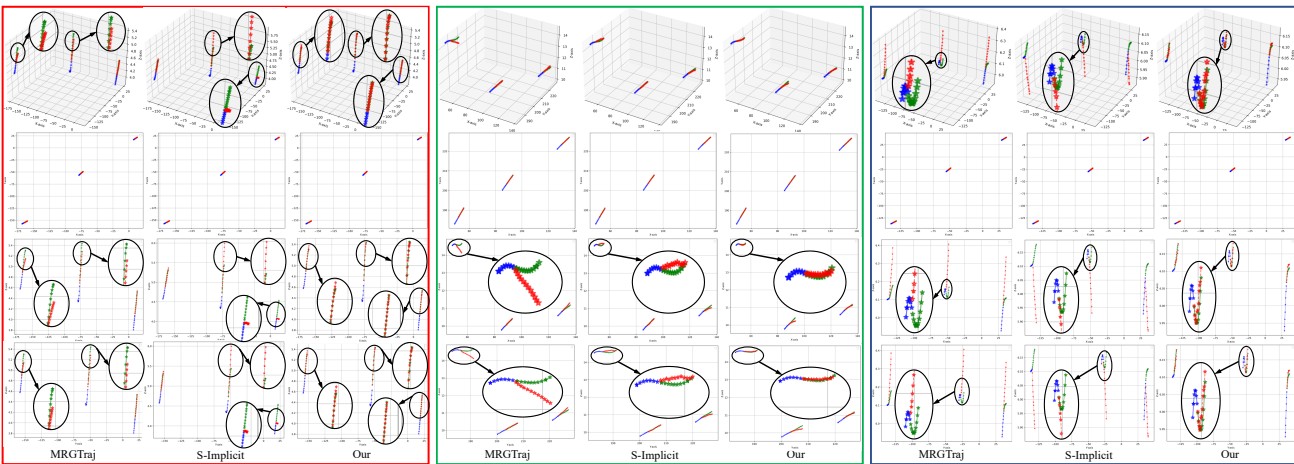

(a) Sample #1: linear trajectories (easy)    (b) Sample #2: curved trajectories (medium)   (c) Sample #3: fluctuating trajectories (hard)

Figure 5: Visual comparisons between our method with MRGTraj and S-Implicit on 3D trajectory prediction using samples featuring linear, curved, and fluctuating trajectories denoting easy, medium, and hard prediction difficulties. The first to the fourth rows are the orthographic projections and front, top, and side views of visualized results. Dotted lines with blue, green, and red colors represent the observed trajectories, the ground-truth future trajectories, and the predicted future trajectories.

where $(\cdot)^T$ denotes transposition, and $\alpha_{c,\nu}$ is the attention weight for the $\nu$-axis. Additionally, the aggregation gate is defined as:

$$g_i^{\nu,c} = \sigma(W_{\nu c}[F_i^{hc,l}; F_i^{h\nu,l}] + b_{\nu c}), \quad \nu \in \{x, y, z\}, \quad (15)$$

where $g_i^{\nu,c}$ represents the gate vector for the $\nu$-axis, while $W_{\nu c}$ and $b_{\nu c}$ are linear transformations and bias terms. After that, the message aggregation function $M_\nu$ is defined as:

$$M_\nu(\hat{hc}_i^{t,l}, \hat{h\nu}_i^{t,l}) = \alpha_{c,\nu} W_{nc} g_i^{\nu,c} \odot \hat{h\nu}_i^{t,l}, \; \nu \in \{x, y, z\}, \quad (16)$$

where $W_{nc}$ is a linear transformation.

Finally, after a round of refinements through the SA and SC modules, the refined cell states are used to update the hidden states via the output gate of the vanilla LSTM, as shown in Equation 30 in the Appendix. Using these updated hidden states, the LSTMs predict the offsets $\Delta\nu_i^f, \nu \in \{x, y, z\}$ to refine the coarse predictions on each axis.

### 4.3. Loss Function

Aside from the specialized loss of the baseline method LBEBM, the loss for our method is defined as follows:

$$
\begin{aligned}
\mathcal{L} &= \frac{1}{N} \sum_{i \in \{1,...,N\}} \sum_{\nu \in \{x,y,z\}} (\mathcal{L}_i^{k,\nu} + \mathcal{L}_i^{f,\nu} + \mathcal{L}_i^{o,\nu}), \\
\mathcal{L}_i^{k,\nu} &= L_2(\nu_i^k, \hat{\nu}_i^k), \quad \mathcal{L}_i^{f,\nu} = L_2(\nu_i^f, \hat{\nu}_i^f), \\
\mathcal{L}_i^{o,\nu} &= L_2(\Delta\nu_i^f, \nu_i^f - \hat{\nu}_i^f).
\end{aligned}
\quad (17)
$$

where $\nu_i^k$ and $\nu_i^f$ denote the ground truth key points and future trajectory on the $\nu$-axis, respectively, and $L_2$ represents the mean squared error.

## 5. Experiments
### 5.1. Implementation Details
Our method uses observed 8 frames to predict future 12 frames. The key points predicted are the 3rd, 6th, 9th, and

| Metric | Iteration Number # | | | | Metric | Layer Number # | | | |
|---|---|---|---|---|---|---|---|---|---|
| | 1 | 2 | 3 | 4 | | 1 | 2 | 3 | 4 |
| ADE | 0.62 | 0.67 | **0.58** | 0.63 | ADE | **0.58** | 0.64 | 0.64 | 0.65 |
| FDE | 1.05 | 1.21 | **1.02** | 1.07 | FDE | **1.02** | 1.07 | 1.06 | 1.10 |

Table 3: Analysis of different iterations of the SC and SA on the 3DMoTraj dataset.

Table 4: Analysis of different layers of the SCA-LSTM on the 3DMoTraj dataset.

| Variant ID | Components | | | | Performance | |
|---|---|---|---|---|---|---|
| | BD | DD | VL | SL | ADE | FDE |
| 1 | ✓ | | | | 0.84 | 1.47 |
| 2 | | ✓ | | | 0.68 | 1.19 |
| 3 | | ✓ | ✓ | | 0.66 | 1.05 |
| 4 | | ✓ | | ✓ | **0.58** | **1.02** |

Table 5: Ablation study on our method using the 3DMoTraj dataset. **BD** refers to our baseline's decoder, **DD** represents our decoupled trajectory prediction part, **VL** stands for the vanilla LSTM, and **SL** refers to our correlated trajectory refinement part, SCA-LSTM.

12th frames of the future trajectory. The primary components of our approach are implemented using MLP or LSTM. Training is conducted with a batch size of 70 for 100 epochs, employing the Adam optimizer with an initial learning rate of 0.0005. The learning rate decayed by 0.5 after the 5th epoch. Results are evaluated using (Average Displacement Error) ADE and (Final Displacement Error) FDE metrics.

### 5.2. Comparisons to State-of-the-art Methods

We present the performance of our proposed method and benchmark it against state-of-the-art methods on the 3DMoTraj dataset, as detailed in Table 2. We modify all compared methods to accept 3D observed trajectories as inputs and predict 3D future trajectories. Additionally, we adjust their data augmentation strategies, such as rotation, flipping, and reversing, to their 3D counterparts for consistency. The experimental results indicate that our method outperforms all evaluated approaches in 3D trajectory prediction. Specifically, our method achieves a 14.6% improvement in the ADE metric, reducing it from 0.68 to 0.58, and a 16.3% enhancement in the FDE metric, lowering it from 1.22 to 1.02, compared to the second-best method, S-Implicit. While NPSN shows an FDE metric close to ours, our ADE metric surpasses it by 22.7%, decreasing from 0.75 to 0.58.

Furthermore, we visualize some samples compared to leading methods MRGTraj and S-Implicit, as shown in Figure 5. The three visualized samples illustrate the ability of these methods to predict linear, curved, and fluctuating trajectories with increasing difficulty levels. The visual results indicate: 1) all compared methods perform well on the x- and y-axes, reflecting their original 2D design; 2) our method surpasses MRGTraj and S-Implicit in 3D trajectory prediction, particularly along the z-axis; and 3) sample #3 validates our method's marked improvement in predicting complex, fluc-

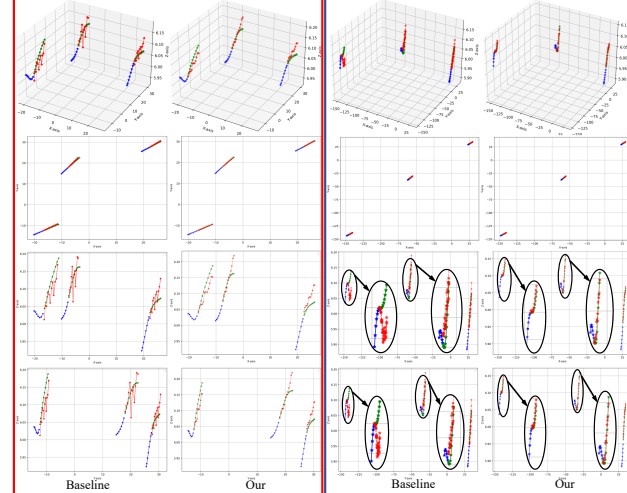

Figure 6: Visual comparisons before and after applying our proposed components on the baseline method. Rows one to four denote different views of the visualized results.

tuating trajectories, which pose a significant challenge for methods designed for 2D prediction.

### 5.3. Hyperparameter Analysis

Training our proposed method involves two key hyperparameters. One is the iterations of the SC and SA modules, and the other is the layers of the SAC-LSTM. This subsection analyzes them one after another.

**Iterations of the SC and SA modules.** Each iteration of the SC and SA modules refines the cell and hidden states of the LSTM. We conduct experiments with counts of $\{1, 2, 3, 4\}$ to choose the optimal number of iterations. The SAC-LSTM layer is fixed at 1 in this experiment to eliminate confounding variables. As shown in Table 3, our method performs best when the iteration number is set to 3.

**Layers of the SCA-LSTM.** When utilizing multiple SCA-LSTM layers, each layer refines the previous layer's output and serves as input for the subsequent one. Different numbers of SAC-LSTM layers can produce varying outcomes. To evaluate the effect of layer depth, we experiment with SAC-LSTM layers of $\{1, 2, 3, 4\}$, as shown in Table 4. The iteration count for the SC and SA modules is fixed at 3, as determined from the previous experiment. The experimental results confirm that using a single SAC-LSTM layer achieves the best performance.

### 5.4. Ablation Study and Discussion

We conduct an ablation study to evaluate our method's core components, *i.e.*, decoupled trajectory prediction and correlated trajectory refinement. The results, presented in Table 5, reveal several vital insights. First, variants 1 and 2 reveal that the decoupled trajectory prediction strategy outperforms the LBEBM decoder by 19.0%/19.0% in ADE/FDE, highlighting its effectiveness. Second, variants 2 and 3 indicate that using vanilla LSTM for refinement provides a performance

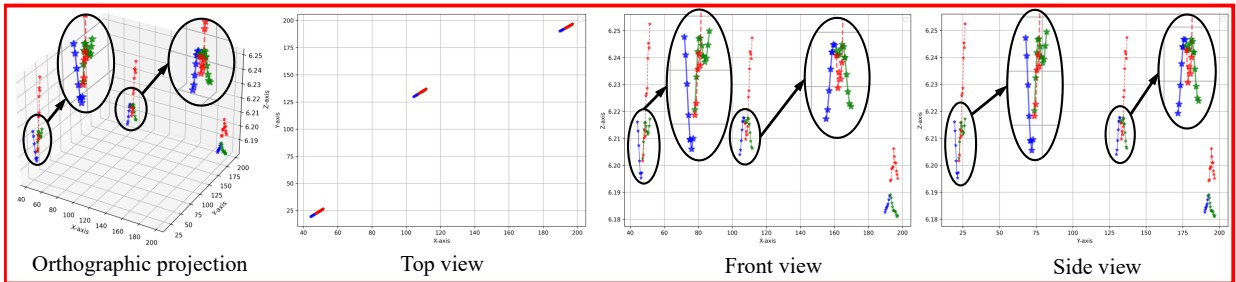

Figure 7: A representative failure case generated by our method on the 3DMoTraj dataset.

| Dataset | ETH/UCY | | SDD | |
|---|---|---|---|---|
| Method | BL | Our | BL | Our |
| ADE | 0.22 | 0.21 | 9.20 | 8.98 |
| FDE | 0.40 | 0.38 | 16.47 | 15.93 |

Table 6: Discussion of the generalization ability of our method for 2D trajectory prediction using the ETH/UCY and SDD datasets. **BL** refers to the baseline LBEBM.

boost of 2.9%/11.8%, confirming that trajectory refinement can further improve prediction accuracy. Finally, variants 3 and 4 show that our proposed SCA-LSTM outperforms the vanilla LSTM by 12.1%/2.9%, validating that SCA-LSTM, which models inter-axis correlations, significantly improves trajectory refinement.

We also visualize two typical samples before and after applying our method, as shown in Figure 6. While the baseline performs well in predicting trajectories along the x- and y-axes, it struggles with the additional z-axis of 3D trajectories. Our method enhances the baseline's performance, particularly in predicting the z-axis component, further demonstrating the effectiveness of our core components.

Furthermore, to validate the generalization ability of the proposed method for 2D trajectory prediction, we conduct experiments on the pedestrian trajectory datasets ETH/UCY and SDD, as shown in Table 6. Our method outperforms the baseline by 4.5% and 5.0% on the ETH/UCY dataset and by 2.4% and 3.3% on the SDD dataset. These experimental results demonstrate that our decoupled prediction strategy can also improve the prediction accuracy of 2D trajectory.

### 5.5. Model Efficiency Analysis

We analyze the model efficiency of the proposed method by comparing its parameters, computational cost, and run-time performance with several state-of-the-art methods, as illustrated in Table 7. All models in this experiment are tested on an NVIDIA 2080 Ti GPU using an input size of $70 \times 8 \times 3$, where 70 represents the number of agents predicted simultaneously-exceeding the agent count in most real-world applications. The results demonstrate that our method achieves the best performance with a relatively good model efficiency, making it suitable for deployment on embedded robotic systems. Additionally, with an inference speed exceeding 12 FPS, our method meets the real-time

| Method | Paramters (M) | FLOPs (G) | Speed (s) | ADE/FDE |
|---|---|---|---|---|
| MSRL | 0.59 | 0.12 | 0.09 | 1.50 / 2.13 |
| LBEBM | 1.24 | 0.09 | 0.05 | 0.84 / 1.47 |
| NPSN | 0.22 | 0.14 | 1.29 | 0.75 / 1.04 |
| CausalHTP | 0.04 | 0.16 | 2.54 | 0.71 / 1.30 |
| MRGTraj | 4.36 | 20.04 | 0.06 | 0.69 / 1.36 |
| Our | 3.41 | 0.24 | 0.08 | 0.58 / 1.02 |

Table 7: Comparison with state-of-the-art methods in terms of parameters, computational costs, and inference speed on the 3DMoTraj dataset. All methods are tested using an NVIDIA 2080 Ti GPU with an input size of $70 \times 8 \times 3$.

decision-making requirements of robotics applications.

### 5.6. Failure Case Analysis

We present a representative failure case in Figure 7, which illustrates that our method struggles when predicting trajectories with multiple sharp bends in short time intervals. Incorporating more advanced interaction modeling techniques or integrating 3D point cloud maps could provide additional structural constraints, thereby improving this case. Nevertheless, as our primary focus is reducing the prediction complexity of 3D trajectories, we adopt a simple interaction modeling strategy and do not use 3D point cloud maps, leading to suboptimal performance in this case. Exploring stronger interaction modeling methods and incorporating 3D environmental representations will be important directions for future work.

## 6. Conclusion

In this paper, we present a large-scale 3D trajectory dataset, 3DMoTraj, which includes frame-wise annotations for both static and dynamic intentions. This dataset facilitates research on 3D trajectory prediction. Additionally, we propose a novel 3D trajectory prediction method that addresses the increased prediction complexity encountered when moving from 2D to 3D trajectories. Experiments on the 3DMoTraj dataset demonstrate that our method outperforms all state-of-the-art models tested, validating its effectiveness in reducing prediction complexity and enhancing prediction accuracy in 3D. In future work, we aim to collect a 3D trajectory dataset of unmanned aerial vehicles (UAVs) operating in air environments to further validate our method and advance research in 3D trajectory prediction.

## Acknowledgements

This work is supported in part by the National Natural Science Foundation of China under grant No. 62202308, in part by the Guangdong Research Team for Communication and Sensing Integrated with Intelligent Computing under project No. 2024KCXTD047, in part by the Youth Innovation Promotion Association CAS under grant No. 2022133, in part by the Science and Technology Commission of Shanghai under grant No. 24ZR1400400, and in part by the Institute of Information & Communications Technology Planning & Evaluation (IITP) grant funded by the Korean Government (MSIT) (No. RS-2024-00457882, National AI Research Lab Project).

## Impact Statement

The work presented in this paper aims to advance the field of 3D trajectory prediction. Our work may have potential societal impacts in various industries, including autonomous driving, drone operations, and robotics. While these impacts are still in their early stages, we believe that there is no need to specifically highlight any particular ethical issues or societal consequences at this point.

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

This supplementary file provides more details to demonstrate the benefits of our proposed dataset and method.

- Prediction complexity analysis.
- Details of vanilla LSTM.
- The motion trajectory visualization.
- The distance distributions for different scenarios.
- The velocity distributions for different scenarios
- The acceleration distributions for different scenarios.
- The curvature distributions for different scenarios.

## A. Prediction Complexity Analysis

The prediction complexity increases nearly twofold when transitioning from 2D $(x, y)$ to 3D $(x, y, z)$ trajectories. Predicting 2D trajectory is equivalent to optimizing 2D Gaussian distributions to approximate the possible locations of each 2D point:

$$f(x, y) = \frac{1}{2\pi \left|\Sigma_{xy}\right|^{1/2}} \exp\left(-\frac{1}{2} \begin{bmatrix} x - \mu_x & y - \mu_y \end{bmatrix} \Sigma_{xy}^{-1} \begin{bmatrix} x - \mu_x \\ y - \mu_y \end{bmatrix}\right), \tag{18}$$

where $\Sigma_{xy}$ denotes the covariance matrix for the 2D Gaussian distribution and can be represented as follows:

$$\Sigma_{xy} = \begin{pmatrix} \sigma_x^2 & \rho_{xy}\sigma_x\sigma_y \\ \rho_{xy}\sigma_x\sigma_y & \sigma_y^2 \end{pmatrix}. \tag{19}$$

Therefore, the prediction of each 2D point involves optimizing 5 free parameters $\left[\mu_x, \mu_y, \sigma_x^2, \sigma_y^2, \rho_{xy}\right]$. Similarly, predicting a 3D point is equivalent to optimizing a 3D Gaussian distribution:

$$f(x, y, z) = \frac{1}{(2\pi)^{3/2} \left|\Sigma_{xyz}\right|^{1/2}} \exp\left(-\frac{1}{2} \begin{bmatrix} x - \mu_x & y - \mu_y & z - \mu_z \end{bmatrix} \Sigma_{xyz}^{-1} \begin{bmatrix} x - \mu_x \\ y - \mu_y \\ z - \mu_z \end{bmatrix}\right), \tag{20}$$

where $\Sigma_{xyz}$ is the covariance matrix for the 3D Gaussian distribution and can be represented as follows:

$$\Sigma_{xyz} = \begin{pmatrix} \sigma_x^2 & \rho_{xy}\sigma_x\sigma_y & \rho_{xz}\sigma_x\sigma_z \\ \rho_{xy}\sigma_x\sigma_y & \sigma_y^2 & \rho_{yz}\sigma_y\sigma_z \\ \rho_{xz}\sigma_x\sigma_z & \rho_{yz}\sigma_y\sigma_z & \sigma_z^2 \end{pmatrix}. \tag{21}$$

This implies that there are 9 parameters $\left[\mu_x, \mu_y, \mu_z, \sigma_x^2, \sigma_y^2, \sigma_z^2, \rho_{xy}, \rho_{yz}, \rho_{xz}\right]$ to be optimized for predicting each 3D point. Consequently, the complexity of predicting one 3D point is approximately twice that of one 2D point.

For a 3D Gaussian distribution, by substituting the covariance matrix $\Sigma_{xyz}$ into Equation 20, we obtain its expanded form:

$$f(x, y, z) = \frac{1}{(2\pi)^{3/2} \left|\Sigma_{xyz}\right|^{1/2}} \exp\left(-\frac{1}{2}\left(\frac{(x-\mu_x)^2}{\sigma_x^2} + \frac{(y-\mu_y)^2}{\sigma_y^2} + \frac{(z-\mu_z)^2}{\sigma_z^2}\right.\right.$$
$$\left.\left. -2\rho_{xy}\frac{(x-\mu_x)(y-\mu_y)}{\sigma_x\sigma_y} - 2\rho_{xz}\frac{(x-\mu_x)(z-\mu_z)}{\sigma_x\sigma_z} - 2\rho_{yz}\frac{(y-\mu_y)(z-\mu_z)}{\sigma_y\sigma_z}\right)\right). \tag{22}$$

The exponential term of Equation 22 can be decomposed into two multiplicative components, such that Equation 22 can be expressed as follows:

$$f(x, y, z) = \underbrace{\frac{1}{(2\pi)^{3/2}\sqrt{\sigma_x^2\sigma_y^2\sigma_z^2}} \exp\left(-\frac{1}{2}\left(\frac{(x-\mu_x)^2}{\sigma_x^2} + \frac{(y-\mu_y)^2}{\sigma_y^2} + \frac{(z-\mu_z)^2}{\sigma_z^2}\right)\right)}_{\text{Part-I}} \times$$

$$\underbrace{\frac{\sqrt{\sigma_x^2\sigma_y^2\sigma_z^2}}{\left|\Sigma_{xyz}\right|^{1/2}} \exp\left(\frac{1}{2}\left(2\rho_{xy}\frac{(x-\mu_x)(y-\mu_y)}{\sigma_x\sigma_y} + 2\rho_{xz}\frac{(x-\mu_x)(z-\mu_z)}{\sigma_x\sigma_z} + 2\rho_{yz}\frac{(y-\mu_y)(z-\mu_z)}{\sigma_y\sigma_z}\right)\right)}_{\text{Part-II}}, \tag{23}$$

where $\mathrm{Part-I}$ and $\mathrm{Part-II}$ denote the independent and correlation parts, respectively. $\mathrm{Part-I}$ corresponds to the distribution of each variable within its own dimension, while $\mathrm{Part-II}$ captures the interdependencies between variables in the 3D Gaussian distribution. This decomposition underscores that the overall Gaussian distribution is the product of independent Gaussian distributions ($\mathrm{Part-I}$) and a correction factor that accounts for inter-axis correlations ($\mathrm{Part-II}$). Additionally, the independent part can further be expressed as the product of three 1D Gaussian distributions:

$$\mathrm{Part-I} = \prod_{\nu \in \{x,y,z\}} \left( \frac{1}{\sqrt{2\pi\sigma_\nu^2}} \exp\left(-\frac{(x_\nu - \mu_\nu)^2}{2\sigma_\nu^2}\right) \right). \tag{24}$$

Therefore, we propose a divide-and-conquer strategy for predicting 3D trajectories. This approach first decouples the prediction process along three independent axes, mitigating the impact of inter-axis correlations. Subsequently, the inter-axis correlations are modeled to refine the predictions. In this way, the prediction of each 3D point with 9 parameters is first simplified to optimize three 1D Gaussian distributions with 6 free parameters, thereby reducing the overall prediction complexity.

## B. Details of Vanilla LSTM

LSTM is a Recurrent Neural Network (RNN) designed to process sequential data such as time series, natural language text, etc. Traditional RNNs struggle with long sequences due to the vanishing or exploding gradient problem. LSTM addresses this by incorporating a gating mechanism that helps capture long-term dependencies effectively.

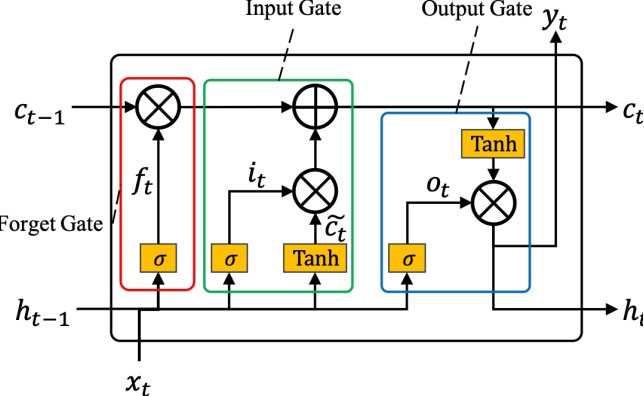

Figure 8: Illustration of LSTM cell.

Vanilla LSTM refers to the basic form of LSTM, composed of a forget gate, input gate, output gate, and cell state, as illustrated in Figure 8. At each time step $t$, the LSTM receives an input vector $x_t$, a previous hidden state $h_{t-1}$, and a previous cell state $c_{t-1}$. Then the four modules compute as follows:

**Forget Gate:** The forget gate decides what information to remove from the cell state. It takes the previous hidden state $h_{t-1}$ and current input $x_t$, and computes a forget ratio:

$$f_t = \sigma(W_f \cdot [h_{t-1}, x_t] + b_f), \tag{25}$$

where:

- $f_t$: Forget gate activation vector at time $t$,
- $\sigma$: Sigmoid activation function,
- $W_f$: Weight matrix for the forget gate,
- $b_f$: Bias vector for the forget gate,
- $h_{t-1}$: Hidden state from the previous time step,
- $x_t$: Input vector at the current time step.

**Input Gate:** The input gate determines what new information to add to the cell state. The gate activation is computed as:

$$i_t = \sigma(W_i \cdot [h_{t-1}, x_t] + b_i), \tag{26}$$

and the candidate memory content is:

$$\tilde{c}_t = \tanh(W_c \cdot [h_{t-1}, x_t] + b_c), \tag{27}$$

where:

- $i_t$: Input gate activation vector,
- $\tilde{c}_t$: Candidate cell state vector,
- $W_i, W_c$: Weight matrices for the input gate and cell state,
- $b_i, b_c$: Bias vectors for the input gate and cell state,
- $\tanh$: Hyperbolic tangent activation function.

**Cell State Update:** The cell state is updated using information from the forget and input gates:

$$c_t = f_t \odot c_{t-1} + i_t \odot \tilde{c}_t, \tag{28}$$

where:

- $c_t$: Cell state vector at time $t$,
- $c_{t-1}$: Cell state vector from the previous time step,
- $\odot$: Element-wise multiplication.

**Output Gate:** The output gate decides the current hidden state (and the output of the LSTM). The gate activation is:

$$o_t = \sigma(W_o \cdot [h_{t-1}, x_t] + b_o), \tag{29}$$

and the hidden state is computed as:

$$h_t = o_t \odot \tanh(c_t), \tag{30}$$

where:

- $o_t$: Output gate activation vector,
- $h_t$: Hidden state vector at time $t$,
- $W_o$: Weight matrix for the output gate,
- $b_o$: Bias vector for the output gate.

In summary, the LSTM performs the following operations at each time step:

1. Inputs the previous hidden state $h_{t-1}$ and current input $x_t$.
2. Uses the forget gate to decide which past information to discard.
3. Updates the cell state with the input gate and candidate memory.
4. Computes the current hidden state using the output gate.

## C. The Motion Trajectory Visualization

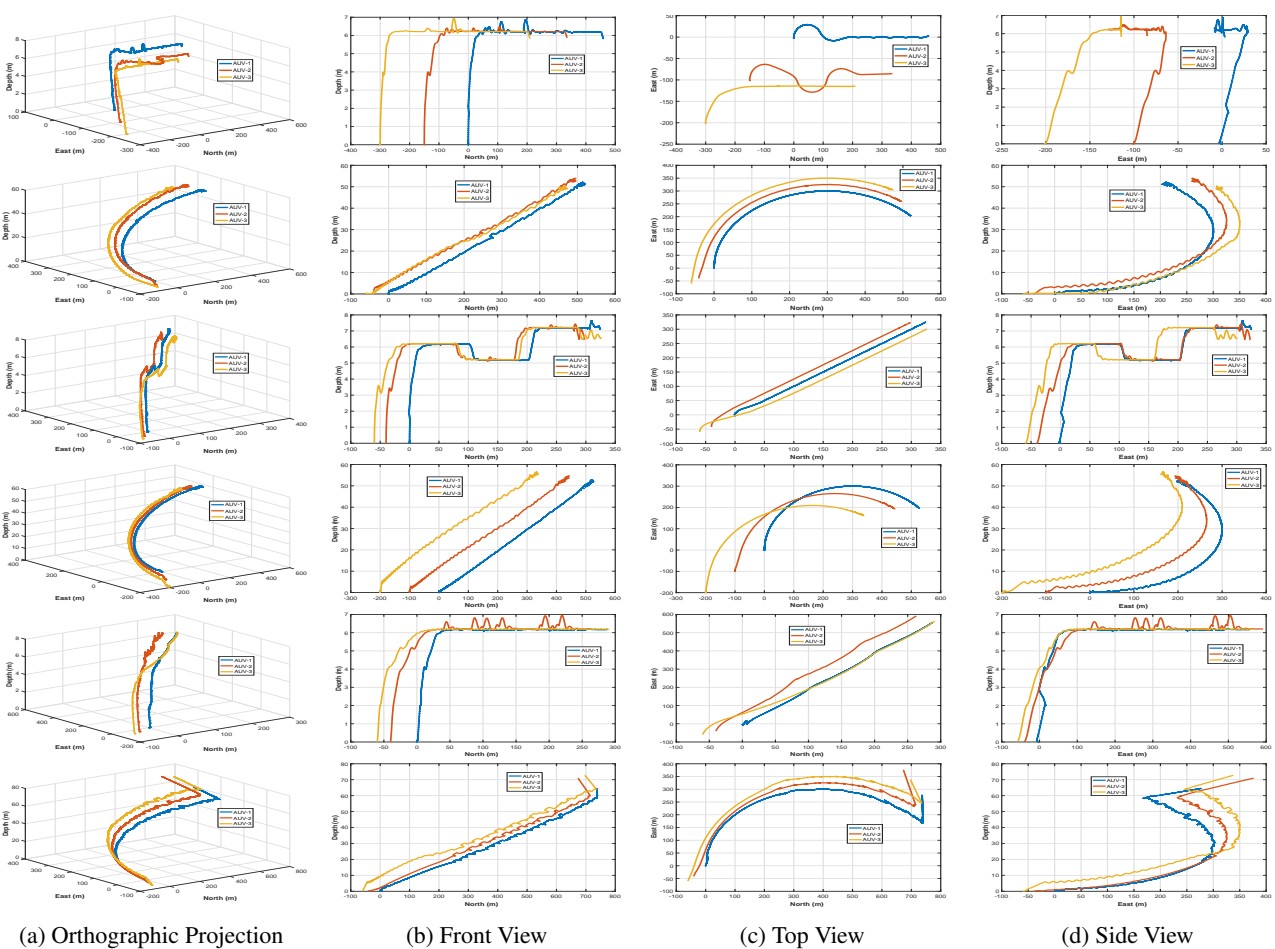

(a) Orthographic Projection      (b) Front View      (c) Top View      (d) Side View

Figure 9: Orthographic projections, along with front, top, and side views of 3D trajectories for the remaining five underwater scenarios in the 3DMoTraj dataset.

## D. The Distance Distributions for Different Scenarios

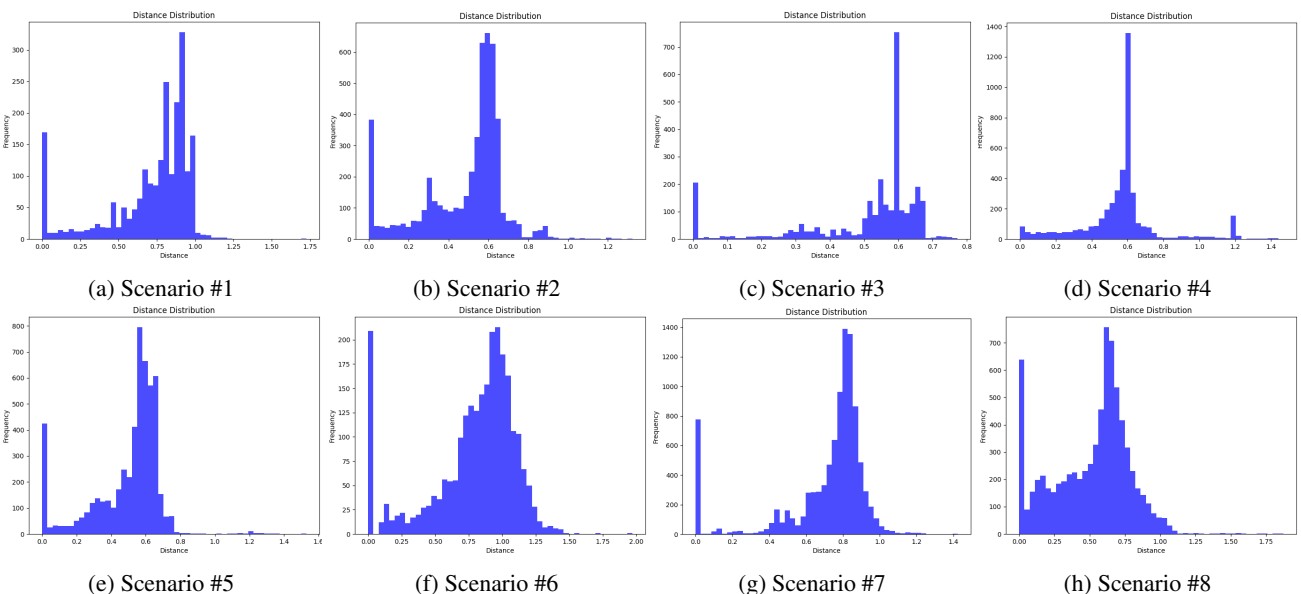

Figure 10: The distance distributions for scenarios #1 through #8.

## E. The Velocity Distributions for Different Scenarios

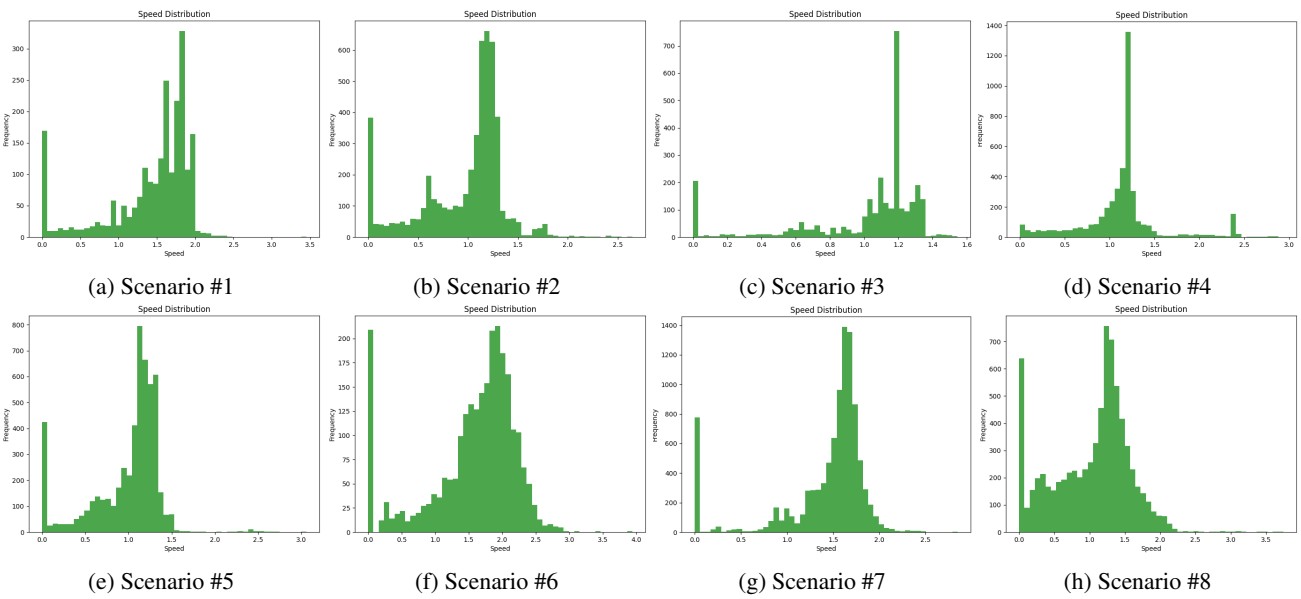

Figure 11: The velocity distributions for scenarios #1 through #8.

## F. The Acceleration Distributions for Different Scenarios

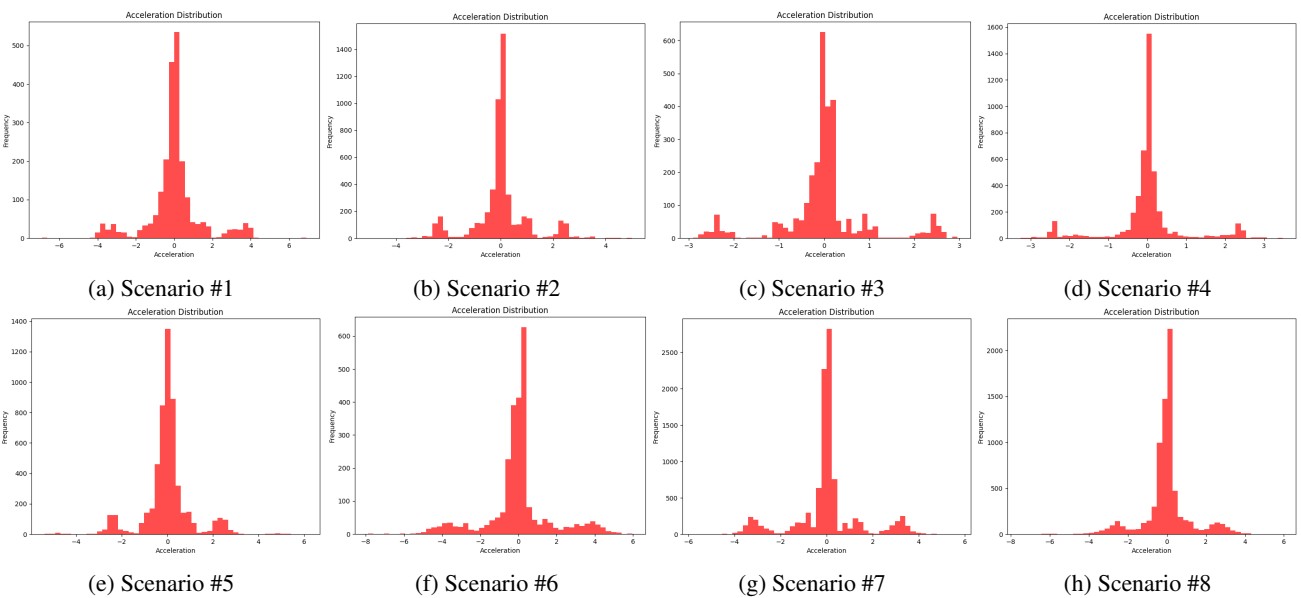

Figure 12: The acceleration distributions for scenarios #1 through #8.

## G. The Curvature Distributions for Different Scenarios

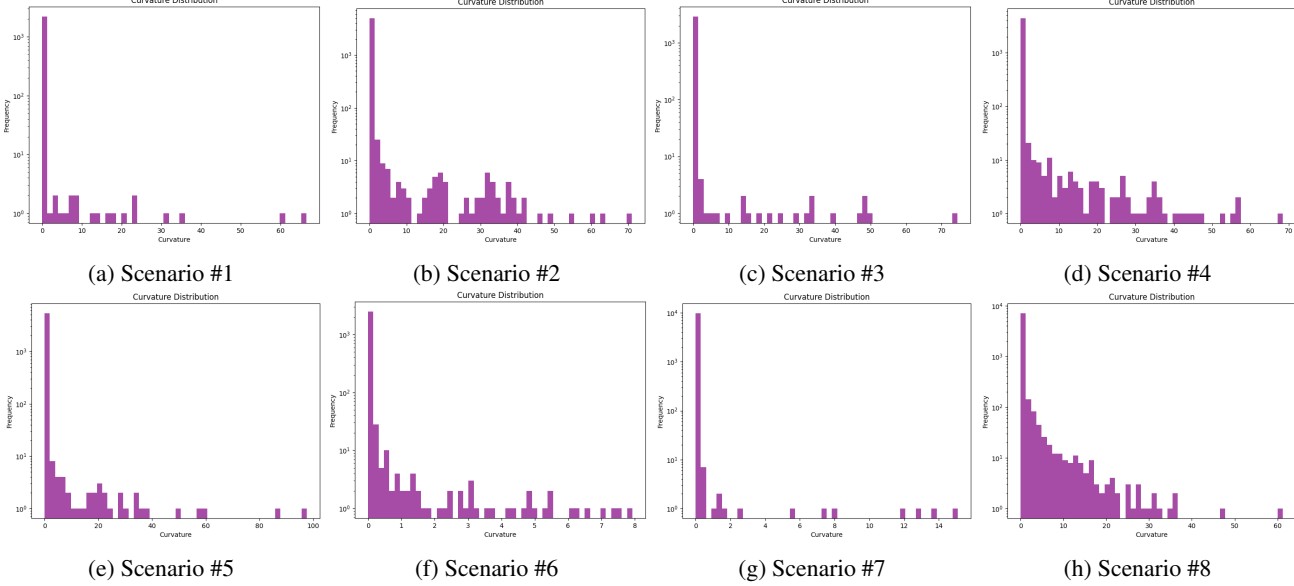

Figure 13: The curvature distributions for scenarios #1 through #8.

