# OpenReview forum: "Three-Dimensional Trajectory Prediction with 3DMoTraj Dataset"
_ICML.cc/2025/Conference — ICML 2025 poster_

### Official Review · Reviewer_Z4gj · 2025-03-07

**Overall Recommendation:** 3

**Summary:**

In this paper, the authors address the challenge of predicting 3D trajectories, which is more complex than 2D trajectory prediction. To achieve this goal, the authors first introduce the 3DMoTraj dataset, collected from unmanned underwater vehicles (UUVs) in oceanic environments. Then, they propose a new method with two key components: decoupled trajectory prediction and correlated trajectory refinement. Based on the new dataset and solution, they reported extensive experiments to show the solution’s superior performance in 3D trajectory prediction.

## Update After Rebuttal
The authors responded to my questions in a great way. They addressed my concerns. I do not have additional questions. I may keep my rating unchanged for this paper (weak accept) for the following reasons:
1. The newly collected dataset is they key factor. Collecting more data with annotations is a good contribution to the community. Thus, I prefer to accept this paper.
2.  However, the improved accuracy claim in the paper is not that convincing, which resulting in a "weak accept" score.

**Claims And Evidence:**

Overall, claims made in this paper are supported by evidence.
1. The dataset. This paper provides detailed description of the dataset collection, including basic, motion, curvature, intention information. Besides that, the authors also report statistics information about the dataset.
2. The proposed solution has a detailed description and complexity analysis. The additional information reported in the supplementary is also helpful.

The improved accuracy claim may need a stronger support.
In Table 2, the proposed solution performs better than other existing work on the newly collected dataset, e.g., MRGTraj (2023), MS-TIP(2024). However, in Table 6, on other existing 2D trajectory prediction datasets, only one solution (LBEMB, 2021) was compared, which makes the generalization capability of the proposed solution and accuracy improvement claim become a little bit less convincing.

**Essential References Not Discussed:**

It will be better if the authors could discuss more recent work, for example:
* Lan Feng, Mohammadhossein Bahari, Kaouther Messaoud Ben Amor, Éloi Zablocki, Matthieu Cord, and Alexandre Alahi. "Unitraj: A unified framework for scalable vehicle trajectory prediction." In European Conference on Computer Vision, pp. 106-123. Cham: Springer Nature Switzerland, 2024.
* Yi Xu, and Yun Fu. "Adapting to length shift: Flexilength network for trajectory prediction." In Proceedings of the IEEE/CVF Conference on Computer Vision and Pattern Recognition, pp. 15226-15237. 2024.
* Moein Younesi Heravi, Youjin Jang, Inbae Jeong, and Sajib Sarkar. "Deep learning-based activity-aware 3D human motion trajectory prediction in construction." Expert Systems with Applications 239 (2024): 122423.
* Zhuoyong Shi, Jiandong Zhang, Guoqing Shi, Longmeng Ji, Dinghan Wang, and Yong Wu. "Design of a UAV Trajectory Prediction System Based on Multi-Flight Modes." Drones 8, no. 6 (2024): 255.

**Experimental Designs Or Analyses:**

Overall, the experimental designs and analyses make sense. However, as mentioned above, it will be better to compare the proposed solution with more existing solutions on public datasets to prove the good generalization capability of the proposed solution.

**Methods And Evaluation Criteria:**

The proposed methods and evaluation criteria make sense for the trajectory prediction task. It will be better, if the authors could report the accuracy for each prediction step. In this case, readers may get more sense about the performance of the proposed solution, e.g., the model performs better at the first few steps and the performs worse at the last step, etc.

**Other Comments Or Suggestions:**

1. Maybe we can try GRU instead of LSTM. There are some work show GRUs achieves better resutls than LSTMs.
2. It seems like the content of Section 4.1 does not exactly match with Figure 4. For example, Equation (2) has a concatenation operation, but we cannot find it in Figure 4. Maybe we can refine the content and the figure.
3. Even though the authors provide a complexity analusis, it will be better if the authors could report the profiled results (latency, power, etc.).

**Other Strengths And Weaknesses:**

Other Strengths:
1. The intuitions behind deigns were described in details, e.g., to address the increased prediction complexity of 3D trajectory, decopuled trajectory prediction and correlated trajectory refinement were proposed.
2. The supplementary material provides more details and analysis help readers better understand their work.

Other Weaknesses:
1. It will be better if the authors could report some failure cases. It may more important for the 3D prediction task, because 3D motion has more freedom than 2D motion and has higher possibility to have failure cases. Reporting failure cases will help readers better understand the performance of the proposed solution.

**Questions For Authors:**

1. Do we need to change the values of iteration number and layer number if we apply the solution on different scenes or datasets?

**Relation To Broader Scientific Literature:**

The proposed solution was adopted from the LBEBM method (Pang et al, 2021). The authors replace LBEBM’s decoder with three independent decoders to predict trajectories separately along the x-, y-, and z-axes. In terms of the newly collected dataset, it should be a good addition to the existing datasets, e.g., ETH/UCY, SDD, etc. (especially for the 3D trajectory prediction task).

**Theoretical Claims:**

This paper does not have challenging mathematical proofs or theoretical claims. (1) The proposed Decoupled Trajectory Prediction and Correlated Trajectory Refinement modules should work in theory. (2) The prediction complexity analysis looks convincing.

---

> ### Author Rebuttal · Authors · 2025-04-01
>
> **Q1: Demonstrate the generalization capability to 2D datasets on more baselines.**
>
> **A1**: We evaluated our prediction strategy on four additional baselines using two widely adopted 2D datasets: ETH&UCY and SDD. All models were trained and tested on the same machine for fair comparison.
>
> |Methods|ETH&UCY|SDD|
> |:----|:----|:----|
> |PECNet|0.30/0.48|10.02/15.79|
> |PECNet+our|0.25/0.42|9.45/15.26|
> |NPSN|0.28/0.44|8.56/14.95|
> |NPSN+our|0.25/0.39|8.34/14.36|
> |MSRL|0.20/0.36|8.36/13.85|
> |MSRL+our|0.19/0.34|8.29/13.56|
> |TrajCLIP|0.21/0.35|7.69/13.31|
> |TrajCLIP+our|0.19/0.32|7.63/13.30|
>
> These results demonstrate that our approach consistently enhances the performance of multiple baselines, further validating its generalization capability to 2D trajectory prediction.
>
> ***
> **Q2: Report the accuracy for each prediction step.**
>
> **A2**: The prediction accuracy at each step for the baseline LBEBM and our method is presented in the table below. Additionally, a visualized line chart of these results is provided; Please refer to Figure-2 at the URL <https://anonymous.4open.science/r/ICML_ID4436/README.md>.
>
> |Steps|#1|#2|#3|#4|#5|#6|#7|#8|#9|#10|#11|#12|
> |:----|:----|:----|:----|:----|:----|:----|:----|:----|:----|:----|:----|:----|
> |LBEBM|0.18|0.34|0.47|0.58|0.69|0.79|0.87|0.95|1.06|1.27|1.40|1.47|
> |Our|0.17|0.29|0.35|0.43|0.49|0.55|0.61|0.67|0.72|0.81|0.89|1.02|
>
> The results and their line chart indicate that our method achieves progressively more significant improvements over the baseline as the prediction horizon extends. This highlights our approach’s ability to mitigate the error accumulation problem in trajectory prediction, further validating its capability to reduce prediction complexity.
>
> ***
> **Q3: Discuss more recent work, for example, [1]-[4].**
>
> **A3**: We will incorporate a discussion of the recent works [1]-[4] in our manuscript.
>
> ***
> **Q4: It will be better to report some failure cases.**
>
> **A4:** We visualized a representative failure case in Figure-3 at <https://anonymous.4open.science/r/ICML_ID4436/README.md>, showing that our method struggles with trajectories featuring multiple sharp bends in short time frames. A more advanced interaction modeling could improve such case. However, as our primary focus is reducing the prediction complexity of 3D trajectories, we adopt a simple interaction modeling strategy, leading to suboptimal performance in this case. In the future, we will explore specialized 3D interaction modeling for complex motion patterns.
>
> ***
> **Q5: Maybe we can try GRU instead of LSTM.**
>
> **A5:** We evaluated the impact of replacing LSTM with GRU in our method, and the results are presented below:
>
> | Architecture|ADE|FDE|
> |:----|:----|:----|
> |LSTM|0.58|1.02|
> |GRU|0.59|1.06|
>
> The results indicate that GRU performs slightly worse than LSTM in our method. However, considering GRU’s higher computational efficiency, we recommend using the GRU-based version for low latency or resource-limited scenarios.
>
> ***
> **Q6: The content of Section 4.1 does not precisely match with Figure 4.**
>
> **A6:** Sorry for the typos in Section 4.1; we will refine them to ensure alignment with Figure 4.
>
> ***
> **Q7: Report the profiled results (latency, power, etc.).**
>
> **A7:** We compared our method’s profiled results with several top-notch methods. Specifically, we tested all models on an NVIDIA 2080 Ti GPU using an input size of 70×8x3, where 70 represents agents' number predicted simultaneously—exceeding the agent count in most real-world applications.
>
> |Methods|Parameters (M)|FLOPs (G)|Inference time (s)|ADE|FDE|
> |:----|:----|:----|:----|:----|:----|
> |MSRL|0.59|0.12|0.09|1.50|2.13|
> |LBEBM|1.24|0.09|0.05|0.84|1.47|
> |NPSN|0.22|0.14|1.29|0.75|1.04|
> |CausalHTP|0.04|0.16|2.54|0.71|1.30|
> |MRGTraj|4.36|20.04|0.06|0.69|1.36|
> |Our|3.41|0.24|0.08|0.58|1.02|
>
> The results show that our method achieves the best performance with a relatively good model efficiency, making it suitable for deployment on embedded robotic systems. Additionally, with an inference speed exceeding 12 FPS, our method meets the real-time decision-making requirements of robotics.
>
> ***
> **Q8: Change the iteration and layer numbers on different scenes or datasets?**
>
> **A8:** It is advisable to adjust these hyperparameters for scenes with significant variations, as different environments impact agent movement to varying degrees. For instance, underwater environments introduce more excellent resistance and turbulence than aerial environments, necessitating different hyperparameters for optimal performance.
>
> ***
> **Reference**
>
> [1] Feng L, et al. Unitraj: A unified framework for scalable vehicle trajectory prediction. ECCV. 2024.
>
> [2] Xu Y, et al. Adapting to length shift: Flexilength network for trajectory prediction. CVPR. 2024.
>
> [3] Heravi M Y, et al. Deep learning-based activity-aware 3D human motion trajectory prediction in construction. ESWA. 2024.
>
> [4] Shi Z, et al. Design of a UAV Trajectory Prediction System Based on Multi-Flight Modes. Drones, 2024.

---

> > ### Comment · Reviewer_Z4gj · 2025-04-05
> >
> > Thank you for your responses. The authors responded to my questions in a great way. They addressed my concerns. I do not have additional questions. Thank you so much.

---

> > > ### Author Response · Authors · 2025-04-05
> > >
> > > Thanks for confirming that all the questions have been answered by the rebuttal. As such, we kindly ask you to consider raising the score of the overall recommendation.
> > >
> > > Thank you,

---

### Official Review · Reviewer_cMBX · 2025-03-13

**Overall Recommendation:** 4

**Summary:**

The paper addresses the problem of 3D trajectory prediction by introducing a novel dataset and an innovative prediction framework. Building upon the 3DMoTraj dataset, they propose a dual-component prediction method that decomposes the 3D trajectory prediction task into two stages. The first stage, decoupled trajectory prediction, independently forecasts trajectories along each spatial axis to reduce overall prediction complexity. The second stage, correlated trajectory refinement, models inter-axis dependencies to generate corrective offsets that enhance the initial predictions. Extensive experiments demonstrate the superiority of the proposed approach.

**Claims And Evidence:**

This paper provides sufficient experiments to support their claims.

**Essential References Not Discussed:**

N/A.

**Experimental Designs Or Analyses:**

Yes.

**Methods And Evaluation Criteria:**

The proposed dataset makes sense for the trajectory prediction task.

**Other Comments Or Suggestions:**

N/A.

**Other Strengths And Weaknesses:**

Strengths:

1. The 3DMoTraj dataset provides a valuable benchmark for evaluating 3D trajectory prediction algorithms in realistic settings. Its frame-wise annotations for both static and dynamic intentions offer precise descriptions of motion characteristics in 3D environments. The proposed dataset is likely to facilitate further research in this area.
2. The authors show that the prediction complexity of 3D trajectories is nearly double that of 2D trajectories, given that a 3D Gaussian distribution requires optimizing 9 parameters. The authors demonstrate that a 3D Gaussian distribution can be decomposed into independent 1D Gaussian components along with a correction factor.
3. The proposed methodology is well designed. The divide-and-conquer strategy, comprising decoupled trajectory prediction and correlated trajectory refinement can reduce the overall prediction complexity.
4. Extensive experiments, including ablation studies and comparisons with state-of-the-art methods, reveal that the proposed approach significantly enhances prediction accuracy and robustness.

Weaknesses:
1. Experimental validation using a 3D trajectory dataset collected from UAVs would further strengthen the paper, given the increased complexity of UAV motion trajectories. I understand that collecting and annotating such a dataset requires substantial effort and may be impractical under current time and resource constraints; however, I look forward to seeing future work on UAV trajectory prediction.
2. A thorough analysis of the model's efficiency is necessary. In the context of robotics applications, prediction algorithms must not only ensure accuracy but also achieve high inference speed to enable real-time decision-making. Evaluating the computational cost and runtime performance of the proposed method would strengthen the paper’s contributions.

**Questions For Authors:**

Refer to above sectoins.

**Relation To Broader Scientific Literature:**

N/A.

**Theoretical Claims:**

I have checked their claims.

---

> ### Author Rebuttal · Authors · 2025-04-01
>
> **Q1: I look forward to seeing future validation on the trajectory dataset of unmanned aerial vehicles.**
>
> **A1**: As outlined in our conclusion, future work will involve collecting a large-scale 3D trajectory dataset from unmanned aerial vehicles (UAVs) to validate our proposed methodology further. Specifically, we plan to capture trajectories across nine distinct environments, including large-scale indoor mall scenarios, urban road airspace scenarios, urban low-altitude logistics corridors, dense vegetation agricultural fields, large industrial park settings, cross-sea bridge inspection sites, post-disaster debris zones, open water airspace environments, and wilderness forest environments. For each environment, we aim to collect six hours of UAV trajectory data, allocating three hours for training, two for validation, and one for testing.
>
> At the current stage, we have collected two environments—large-scale indoor mall scenarios and urban road airspace scenarios—though annotations remain incomplete. In the indoor mall scenario, three or four UAVs simulate coordinated formation movement to mimic goods pickup and delivery tasks. In the urban road airspace scenario, four or six UAVs navigate complex intersections to simulate 3D logistics transmission environments. To evaluate our method on a more general 3D dataset, we randomly selected one hour of annotated data from these two scenarios (30 minutes for training, 20 for validation, and 10 for testing) and benchmarked our approach against other state-of-the-art methods.
>
> | Methods| Large-scale indoor mall scenarios| Urban road airspace scenarios |
> |:---- |:---- |:---- |
> | NPSN | 0.72/0.97 | 0.83/1.31 |
> | MRGTraj | 0.60/0.78 | 0.73/1.23 |
> | CausalHTP | 0.54/0.71 | 0.68/1.18 |
> | S-Implicit | 0.52/0.73 | 0.65/1.25 |
> | MS-TIP | 0.47/0.77 | 0.62/1.12 |
> | TrajCLIP | 0.45/0.74 | 0.61/1.13 |
> | Our | 0.32/0.67 | 0.42/0.95 |
>
> The results demonstrate that our method outperforms all investigated approaches, validating its effectiveness in diverse 3D scenarios. Besides, our method performs best at urban road airspace scenarios, demonstrating the method's ability to predict complex non-formation trajectories. Moreover, our approach performs better in UAV-based 3D environments than UUV-based underwater scenarios, further confirming the greater complexity of underwater trajectory prediction due to higher resistance and turbulence. We have also visualized several predicted trajectories in Figure-4 and Figure-5 at the anonymous URL <https://anonymous.4open.science/r/ICML_ID4436/README.md>, further illustrating the ability of our method in general 3D trajectory prediction.
>
> ***
> **Q2: Evaluating the computational cost and runtime performance.**
>
> **A2**: We compared our method’s computational cost, parameter, and runtime performance with several state-of-the-art methods. Specifically, we tested all models on an NVIDIA 2080 Ti GPU using an input size of 70×8x3, where 70 represents the number of agents predicted simultaneously—exceeding the agent count in most real-world applications. The results are presented below:
>
> | Methods | Parameters (M) | FLOPs (G) | Inference time (s) | ADE | FDE |
> |:---- |:---- |:---- |:---- |:---- |:---- |
> | MSRL | 0.59 | 0.12 | 0.09 | 1.50 | 2.13 |
> | LBEBM | 1.24 | 0.09 | 0.05 | 0.84 | 1.47 |
> | NPSN | 0.22 | 0.14 | 1.29 | 0.75 | 1.04 |
> | CausalHTP | 0.04 | 0.16 | 2.54 | 0.71 | 1.30 |
> | MRGTraj | 4.36 | 20.04 | 0.06 | 0.69 | 1.36 |
> | Our | 3.41 | 0.24 | 0.08 | 0.58 | 1.02 |
>
> The results demonstrate that our method achieves the best performance with a relatively good model efficiency, making it suitable for deployment on embedded robotic systems. Additionally, with an inference speed exceeding 12 FPS, our method meets the real-time decision-making requirements of robotics applications.

---

### Official Review · Reviewer_cvmD · 2025-03-13

**Overall Recommendation:** 2

**Summary:**

Firstly, this paper introduces the 3DMoTraj dataset, a novel 3D trajectory dataset collected from unmanned underwater vehicles (UUVs) in oceanic environments. The dataset includes annotations for both static (endpoint octant) and motion (velocity change) intentions. Secondly, to address the increased complexity of 3D trajectory prediction compared to 2D, this paper proposes a method consisting of two components: decoupled trajectory prediction (independently predicting each axis to reduce complexity) and correlated trajectory refinement (modeling inter-axis correlations to refine predictions). The approach leverages LSTM-based modules with state-correlation and aggregation mechanisms. Experiments on the 3DMoTraj dataset demonstrate improvements over state-of-the-art methods, achieving lower Average Displacement Error (ADE) and Final Displacement Error (FDE). This paper also validates the method’s generalization to 2D datasets like ETH/UCY and SDD.

**Claims And Evidence:**

The claims in this paper are not fully supported by clear and convincing evidence for the following reasons:

1. Dataset Limitations: The 3DMoTraj dataset is self-collected and may be specific to the 3D trajectory prediction model presented in this paper. The dataset is an analog simulation data based on human intervention and has not been validated for its application in real-world scenarios.

2. Method Validation: The model proposed in this paper has only been evaluated on the self-constructed dataset, and has not been experimentally validated against state-of-the-art methods on other 3D datasets, so it does not indicate whether the excellent experimental results stem from the design of the model or from the bias of the specific dataset.

3. Application scenario: Most of the trajectories in the real world are 2D, so many studies are based on 2D trajectories. Many 3D trajectories are military datasets, and the application scenarios of this research are limited.

**Essential References Not Discussed:**

None

**Experimental Designs Or Analyses:**

The experimental designs and analyses in this paper contain critical flaws that compromise the validity of the conclusions.

1. Comparison with State-of-the-Art Methods
 This paper modifies 2D trajectory prediction methods (e.g., SSTGCNN, MSRL) to receive 3D trajectory inputs, and also adjusts the data enhancement strategies of these methods, which may affect the original effectiveness of these methods.
 The model proposed in this paper has only been evaluated on the self-constructed dataset, and has not been experimentally validated against state-of-the-art methods on other 3D datasets, so it does not indicate whether the excellent experimental results stem from the design of the model or from the bias of the specific dataset.

2. Ablation Studies
In the ablation experiments, the authors did not discuss the role of the SC module and SA module in State-Correlation and Aggregation LSTM (SCA-LSTM) separately.

3. Generalization to 2D Datasets
Comparison experiments with a single baseline LBEBM on the ETH/UCY and SDD datasets do not prove that the 3D Decoupled-Correlated trajectory prediction strategy proposed in this paper can improve the performance of 2D trajectory prediction.

**Methods And Evaluation Criteria:**

The proposed method and evaluation criteria partially align with the problem of 3D trajectory prediction but suffer from critical limitations that undermine their suitability for broader validation and practical impact. First, the 3DMoTraj dataset is generated specific to this model, and it is not possible to assess the effectiveness of this method as applied in a realistic scenario. Second, the design concept of this model is simple, mainly based on LSTM and Attention mechanism, which is not innovative enough.

**Other Comments Or Suggestions:**

It is recommended that the authors refine the implementation principles of the SA module and SC module in the model framework diagram in Figure 4.

**Other Strengths And Weaknesses:**

Strengths:

1. The 3DMoTraj dataset focus on 3D trajectories of UUVs in oceanic environments, addressing a critical gap in underwater robotics research. Its annotations for motion and static intentions (velocity changes and endpoint octants) provide a unique foundation for intention-conditioned 3D prediction.

Weaknesses:

1. The 3DMoTraj dataset is self-collected and may be specific to the 3D trajectory prediction model presented in this paper. The dataset is an analog simulation data based on human intervention and has not been validated for its application in real-world scenarios.

2. The model proposed in this paper has only been evaluated on the self-constructed dataset, and has not been experimentally validated against state-of-the-art methods on other 3D datasets, so it does not indicate whether the excellent experimental results stem from the design of the model or from the bias of the specific dataset.

**Questions For Authors:**

Method Scalability: How does your method perform in scenarios with more agents, higher noise, or non-formation dynamics (e.g., adversarial UUV interactions)? Have you tested its scalability to such real-world complexities?

**Relation To Broader Scientific Literature:**

The approach of independently predicting each axis (x, y, z) is a very common mathematical idea. This reduces the complexity of optimizing 3D Gaussian distributions, as theoretically justified in the paper.

The SCA-LSTM module’s state-correlation and aggregation mechanisms draw inspiration from attention-based methods (e.g., Trajectron++, Salzmann et al., 2020) and graph neural networks (e.g., STGAT, Huang et al., 2019).

The paper contributes a specialized dataset and a method for UUV trajectory prediction. While its decoupled-correlated framework and intention annotations build on prior work in 2D and 3D prediction, the narrow application scope and lack of cross-domain validation restrict its broader scientific impact. To better align with the literature, future work should validate the method on diverse 3D datasets and compare against 3D-specific baselines.

**Theoretical Claims:**

This paper makes theoretical claims about the complexity of 3D trajectory prediction and the decomposition of 3D Gaussian distributions. After reviewing the mathematical derivations, the following issues are identified:

1. Validity of Parameter Complexity Analysis
This paper correctly argues that 3D trajectory prediction requires optimizing 9 parameters per point (3 means, 3 variances, 3 correlations), compared to 5 parameters for 2D (2 means, 2 variances, 1 correlation). This analysis is mathematically sound.

2. Decomposition of 3D Gaussian Distributions
The decomposition of the 3D Gaussian distribution into independent 1D Gaussians and a correlation correction term (Equations 22–24) is theoretically valid. The authors correctly show that the exponential term can be split into independent (Part-I) and interdependent (Part-II) components. However, the claim that this reduces the total number of parameters from 9 to 6 is misleading. While Part-I corresponds to 6 parameters (3 means, 3 variances), Part-II still requires modeling 3 correlations (ρ_xy, ρ_xz, ρ_yz), totaling 9 parameters. The decoupling strategy simplifies the optimization process by separating independent and correlated components but does not reduce the total number of parameters.

In summary, the proof and application of the mathematical theory in this paper is correct.

---

> ### Author Rebuttal · Authors · 2025-03-31
>
> **Q1: The dataset is simulation data and has not been validated in real-world scenarios.**
>
> **A1**: While our dataset is based on predefined formation trajectories, real-world disturbances naturally cause deviations from planned paths and formation shifts. These deviations reflect real challenges in trajectory prediction. Moreover, most real-world robotic trajectories are pre-programmed rather than spontaneously generated, meaning our dataset aligns with actual robotic motion patterns, making it representative of real-world scenarios.
>
> ***
> **Q2: Validating the proposed method against top-notch methods on other 3D datasets.**
>
> **A2**: It is hard to find publicly available 3D trajectory dataset. We plan to collect a more general 3D trajectory of unmanned aerial vehicles (UAVs) to validate our method. Specifically, we will capture trajectories across nine environments from the civilian field. Currently, we have collected two environments: large-scale indoor mall scenarios and urban road airspace scenarios. We randomly selected one hour of annotated data from these two scenarios and benchmarked our method against top-notch methods. For more experimental details, please see **Q1** of Reviewer cMBX.
>
> |Methods|Large-scale indoor mall scenarios|Urban road airspace scenarios|
> |:----|:----|:----|
> |NPSN|0.72/0.97 |0.83/1.31|
> |MRGTraj|0.60/0.78|0.73/1.23|
> |CausalHTP|0.54/0.71|0.68/1.18|
> |S-Implicit|0.52/0.73|0.65/1.25|
> |MS-TIP|0.47/0.77|0.62/1.12|
> |TrajCLIP|0.45/0.74|0.61/1.13|
> |Our|0.32/0.67|0.42/0.95|
>
> The results show that our method outperforms all others, validating its effectiveness in diverse 3D scenarios. The visualized results in Figure-4 and Figure-5 at URL <https://anonymous.4open.science/r/ICML_ID4436/README.md> further illustrate its general 3D prediction capability.
>
> ***
> **Q3: Many 3D trajectories are military datasets with limited application scenarios.**
>
> **A3**: 3D trajectory prediction has broad civilian applications. Beyond UAV navigation and obstacle avoidance, it supports urban air mobility, precision agriculture, disaster response, and industrial inspections, ensuring safe and efficient operations. These applications extend its impact far beyond military use.
>
> ***
> **Q4: The claim that reduces the total number of parameters is misleading.**
>
> **A4**: Yes, our decoupling strategy simplifies the optimization process but does not reduce optimized parameters. We will clarify this in our paper.
>
> ***
> **Q5: The effects of modifying the data enhancement strategies of compared methods.**
>
> **A5**: To ensure a fair augmentation, we conducted a comparative study of 2D augmentations (rotation, flipping, and translation) versus their 3D counterparts across several top-notch methods.
>
> |Augmentation|2D strategies|3D strategies|
> |:----|:----|:----|
> |CausalHTP|0.71/1.30|0.69/1.27|
> |MS-TIP|0.70/1.31|0.70/1.29|
> |MRGTraj|0.69/1.36|0.68/1.31|
> |S-Implicit|0.68/1.22|0.66/1.20|
> |Our |0.58/1.02|0.47/0.88|
>
> The results show that 3D strategies provide slight improvements on others but significantly enhance our method, suggesting that 3D strategies are particularly beneficial for our model. For fairness, we use standard 2D strategies with 3D inputs for all methods.
>
> ***
> **Q6: Discussion of the SC and SA modules separately.**
>
> **A6**: We conducted separate ablation studies on the SC and SA modules.
>
> | Settings|ADE|FDE|
> |:---- |:---- |:---- |
> |LSTMOnly|0.66|1.05|
> |LSTM+SC|0.64|1.04|
> |LSTM+SA|0.61|1.02|
> |LSTM+SA+SC|0.58|1.02|
>
> The results indicate that removing the SC or SA module degrades the model's performance, highlighting their contributions.
>
> ***
> **Q7: Generalization to 2D Datasets with more baselines.**
>
> **A7**: We evaluated our prediction strategy on four additional baselines using 2D datasets: ETH&UCY and SDD. To ensure a fair comparison, all models were trained and tested on the same machine.
>
> |Methods|ETH&UCY|SDD|
> |:----|:----|:----|
> |PECNet|0.30/0.48|10.02/15.79|
> |PECNet+our|0.25/0.42|9.45/15.26|
> |NPSN|0.28/0.44|8.56/14.95|
> |NPSN+our|0.25/0.39|8.34/14.36|
> |MSRL|0.20/0.36|8.36/13.85|
> |MSRL+our| 0.19/0.34 |  8.29/13.56|
> |TrajCLIP|0.21/0.35|7.69/13.31|
> |TrajCLIP+our|0.19/0.32|7.63/13.30|
>
> These results show that our approach consistently enhances the performance of multiple baselines, further validating its generalization capability to 2D trajectory prediction.
>
> ***
> **Q8: Refine the implementation principles of the SA and SC modules in Figure 4.**
>
> **A8**: We added the implementation details of the SA and SC modules in Figure 4. Please refer to Figure-1 at URL <https://anonymous.4open.science/r/ICML_ID4436/README.md>.
>
> ***
> **Q9: How does your method perform in real-world scenarios?**
>
> **A9**: As discussed in **Q2**, our method has been evaluated in urban road airspace scenarios, where groups of UAVs navigate complex intersections to simulate 3D logistics transmission. These scenarios involve complex interactions. Our method performs best in this environment, demonstrating its scalability to real-world complexities.

---

### Official Review · Reviewer_3PK9 · 2025-03-23

**Overall Recommendation:** 3

**Summary:**

This paper proposes a 3D trajectory dataset named 3DMoTraj collected from unmanned underwater vehicles (UUVs) in ocean environments, which fills the research gap in this field.
Regarding the setting of 3D trajectory prediction, the paper highlights the challenge of computational complexity and provides theoretic proofs.
The paper proposes a decoupled framework to mitigate the computational complexity, experiments demonstrate the superior accuracy of 3D trajectory prediction.

**Claims And Evidence:**

The paper proposes a decoupled framework to mitigate the computational complexity, but it has not provided ablations or method comparisons on inference cost.

**Essential References Not Discussed:**

No.

**Experimental Designs Or Analyses:**

The paper has not provided experiments to support the claims of mitigating the computational complexity of the 3D trajectory prediction task.

**Methods And Evaluation Criteria:**

Yes.

**Other Comments Or Suggestions:**

Please see the weaknesses.

**Other Strengths And Weaknesses:**

Strengths:

1. The paper proposes a large-scale and diverse 3D trajectory dataset collected from unmanned underwater vehicles in ocean environments, which is a good contribution to the related field.

2. The computational complexity challenges discussed in this paper are well-justified.

3. The performance of the proposed decoupled framework surpasses that of several methods in the 3D trajectory prediction problem, and ablations demonstrate that the proposed modules can enhance prediction accuracy.

Weaknesses:

1. The 3DMoTraj dataset is collected in ocean environments. Extending it to general 3D scenarios is difficult due to underwater dynamics, and the paper should elaborate on the limitations of the application scenarios more formally for clarity.

2. Apart from 2D and 3D, the paper has not provided further discussions on the differences between the proposed framework and other trajectory prediction methods. Trajectory refinement is a common practice in trajectory prediction tasks, such as in MTR, QCNet,  SmartRefine, etc.

**Questions For Authors:**

Is there any available 3D trajectory dataset collected from unmanned aerial vehicles? If so, please provide a discussion on it.

**Relation To Broader Scientific Literature:**

The key contributions of the paper are related to previous 2D trajectory prediction datasets and methods, as well as future 3D trajectory prediction tasks.

**Theoretical Claims:**

The theoretical claims that support the key contributions of this paper are correct.

---

> ### Author Rebuttal · Authors · 2025-03-31
>
> **Q1: The paper tries to mitigate the computational complexity but has not provided ablations or method comparisons on inference cost.**
>
> **A1**: At first, we clarify that the complexity our paper aims to mitigate is prediction complexity, which is crucial for optimizing 3D trajectory prediction. However, this is distinct from computational complexity. It is our fault for misunderstanding. To avoid any confusion, we will explicitly define the term of prediction complexity in our paper. Theoretical analysis in the Appendix and ablation studies confirm that our method effectively reduces the prediction complexity of 3D trajectories.
>
> Computational complexity is indeed an important aspect to evaluate the proposed method; therefore, below we compared inference costs across several state-of-the-art methods using an NVIDIA 2080 Ti GPU. To ensure fair comparisons, all tested methods were modified to accept inputs of shape 70×8×3.
>
> | Methods|Parameters (M)|FLOPs (G)|Inference time (s)|ADE|FDE|
> |:----|:----|:----|:----|:----|:----|
> |MSRL|0.59|0.12|0.09|1.50|2.13|
> |LBEBM|1.24|0.09|0.05|0.84|1.47|
> |NPSN|0.22|0.14|1.29|0.75|1.04|
> |CausalHTP|0.04|0.16|2.54|0.71|1.30|
> |MRGTraj|4.36|20.04|0.06|0.69|1.36|
> |Our|3.41|0.24|0.08|0.58|1.02|
>
> These results show that our method performs best while maintaining relatively high model efficiency, making it suitable for deployment on embedded robotic systems.
>
> ***
> **Q2: Extending 3DMoTraj dataset to general 3D scenarios is difficult due to underwater dynamics.**
>
> **A2**: From the methodological perspective, our motivation is to reduce the prediction complexity of 3D trajectory in a manner that ensures robustness across diverse scenarios. The 3DMoTraj dataset is only one of the validation scenes for our method.
>
> To further establish its generalizability, we are constructing a large-scale 3D trajectory dataset for unmanned aerial vehicles (UAVs). Preliminary results from this dataset confirm that our method extends effectively to general 3D trajectory prediction. Please refer to **Q4** below for more details.
>
> ***
> **Q3: Further discussions on trajectory refinement works, such as MTR, QCNet, SmartRefine, etc.**
>
> **A3**: We modified the trajectory refinement components of MTR [1], QCNet [2], and SmartRefine [3] by replacing them with the refinement module in our method.
>
> |Settings|ADE|FDE|
> |:----|:----|:----|
> |Decoupled prediction+MTR|0.69|1.15|
> |Decoupled prediction+QCNet|0.64|1.04|
> |Decoupled prediction+SmartRefine|0.61|1.05|
> |Decoupled prediction+SCA-LSTM (Our)|0.58|1.02|
>
> The results show that our SCA-LSTM performs best in refining initial predictions. This outcome is expected, as SCA-LSTM explicitly models the inter-axis correlations of predicted trajectories, which are intentionally ignored in the decoupled prediction stage to simplify optimization. In contrast, the refinement modules of MTR, QCNet, and SmartRefine focus on modeling interactions between predictions, global intentions, and local maps for refinement.
>
> We will further elaborate on the differences between our method and other 2D refinement-based methods in our paper.
>
> ***
> **Q4: Provide a discussion on 3D trajectory dataset collected from unmanned aerial vehicles.**
>
> **A4**: As outlined in our conclusion, future work will involve collecting a large-scale 3D trajectory dataset from unmanned aerial vehicles (UAVs) to validate our method further. Specifically, we plan to capture trajectories across nine distinct environments. For each environment, we aim to collect six hours of trajectories, allocating three hours for training, two for validation, and one for testing.
>
> At the current stage, we have collected two environments—large-scale indoor mall scenarios and urban road airspace scenarios—though annotations remain incomplete. We randomly selected one hour of annotated data from these two scenarios (30 minutes for training, 20 for validation, and 10 for testing) and benchmarked our approach against other state-of-the-art methods.
>
> |Methods|Large-scale indoor mall scenarios|Urban road airspace scenarios|
> |:----|:----|:----|
> |NPSN|0.72/0.97|0.83/1.31|
> |MRGTraj|0.60/0.78|0.73/1.23|
> |CausalHTP|0.54/0.71|0.68/1.18|
> |S-Implicit|0.52/0.73|0.65/1.25|
> |MS-TIP|0.47/0.77|0.62/1.12|
> |TrajCLIP|0.45/0.74|0.61/1.13|
> |Our|0.32/0.67|0.42/0.95|
>
> The results demonstrate that our method outperforms all the competitors, validating its effectiveness in diverse 3D scenarios. We have also visualized several predicted trajectories in Figure-4 and Figure-5 at the anonymous URL <https://anonymous.4open.science/r/ICML_ID4436/README.md>, further illustrating the ability of our method in general 3D trajectory prediction.
>
> ***
> **Reference**
>
> [1] Shi S, et al. Motion transformer with global intention localization and local movement refinement. NeurIPS, 2022.
>
> [2] Zhou Z, et al. Query-centric trajectory prediction. CVPR. 2023.
>
> [3] Zhou Y, et al. Smartrefine: A scenario-adaptive refinement framework for efficient motion prediction. CVPR. 2024.

---

> > ### Comment · Reviewer_3PK9 · 2025-04-05
> >
> > After considering the authors' response and other reviewers' comments, I acknowledge that leveraging customized modules to represent intra-axis features and inter-axis correlations respectively could be a potential way to simplify the optimization process of modeling the Gaussian distribution for future trajectories. The authors have provided theoretical analysis and experiments in 2D/3D scenarios to support the method's effectiveness.
> >
> > My remaining concerns are as follows:
> >
> > 1. In the experiments on the ETH&UCY and SDD datasets, the authors compare the baselines with their combinations involving the proposed components. For a fair comparison, it is necessary to also provide an analysis of model parameters and computational complexity.
> >
> > 2. Regarding the proposed 3DMoTraj dataset, I appreciate the authors' effort to collect a more diverse 3D trajectory dataset in the future. However, if the additional data will not be included in the current work, it would be beneficial to indicate the UUV scenario more formally (such as in the dataset's name) for clarity.
> >
> > --Post rebuttal: Thank you for your response. Most of my concerns have been addressed, and I believe this work will provide a valuable contribution to the community. Therefore, I will raise my rating.

---

> > > ### Author Response · Authors · 2025-04-07
> > >
> > > # To Reviewer 3PK9 / follow-up comments
> > > Dear Reviewer 3PK9,
> > >
> > > We sincerely appreciate your follow-up comments to help us improve our work. Accordingly, we have responded to each comment as follows.
> > >
> > > ***
> > > **Q1: For a fair comparison, it is necessary to also provide an analysis of model parameters and computational complexity for experiment on 2D datasets.**
> > >
> > > **A1**: Following your suggestion, we added an analysis of model parameters and computational complexity for our comparative experiments on the 2D datasets ETH&UCY and SDD. All models are trained and tested using the same machine equipped with an NVIDIA 2080 Ti GPU, and modified to accept an input shape of 70×8×3. The results are presented below.
> > >
> > > |Methods|Parameters (M)|FLOPs (G)|Inference time (s)|ETH&UCY (ADE/FDE)|SDD (ADE/FDE)|
> > > |:----|:----|:----|:----|:----|:----|
> > > |PECNet|1.23|0.12|0.03|0.30/0.48|10.02/15.79|
> > > |PECNet + our|2.31|0.18|0.05|0.25/0.42|9.45/15.26|
> > > |NPSN|0.22|0.14|1.26|0.28/0.44|8.56/14.95|
> > > |NPSN + our|1.19|0.21|1.32|0.25/0.39|8.34/14.36|
> > > |LBEBM|1.24|0.09|0.05|0.22/0.40|9.20/16.47|
> > > |LBEBM + our|2.32|0.16|0.07|0.21/0.38|8.98/15.93|
> > > |MSRL|0.59|0.12|0.09|0.20/0.36|8.36/13.85 |
> > > |MSRL + our|1.47|0.17|0.12|0.19/0.34|8.29/13.56|
> > > |TrajCLIP|14.94|18.96|0.28|0.21/0.35|7.69/13.31|
> > > |TrajCLIP + our|16.05|19.02|0.31|0.19/0.32|7.63/13.30|
> > >
> > > The results show that integrating our decoupled trajectory prediction and correlated trajectory refinement introduces minimal overhead among all the methods. On average, it results in an increase of approximately 1M parameters, 0.05-0.08 GFLOPs, and 0.02-0.06 seconds in inference time.
> > >
> > > These results further validate the efficiency and effectiveness of our method's main components, even in general 2D scenarios—making it practical for deployment on various hardware platforms.
> > >
> > > ***
> > > **Q2: It would be beneficial to indicate the UUV scenario more formally (such as in the dataset's name) for clarity.**
> > >
> > > **A2**: The 3DMoTraj dataset primarily focuses on 3D trajectories of UUV operating in oceanic environments. To enhance clarity, we will rename the dataset to 3DMoTraj-UUV to explicitly indicate the UUV scenario. Since accurate 3D trajectory prediction is critical for UUVs, enabling efficient path planning, real-time coordination, and robust obstacle avoidance, we believe that the 3DMoTraj-UUV dataset will be a valuable resource for advancing research in marine robotics and related fields.
> > >
> > > Inspired by your suggestion, for the unmanned aerial vehicles (UAV) datasets currently under collection, we will name them according to their collected scenarios. These datasets will be utilized to further evaluate the generalization ability of our 3D trajectory prediction method in our future work. We would also like to clarify that the UAV data will not be included in the current paper, as they are still in the early stages of collection and not yet ready for release.
> > >
> > >
> > > ***
> > > Thank you once again for your helpful feedback. We hope our response addresses your concern.

---

### Decision · Program_Chairs · 2025-05-01

**Decision:**

Accept (poster)

**Comment:**

This paper was reviewed by four experts in the field, and received three positive ratings (4, 3, 3) and one negative (2). The positive reviews commend the paper's clear justification, the value of the dataset, and the effectiveness of the proposed trajectory prediction method. The negative review appears to be based in misunderstanding. The reviewer states "The dataset is an analog simulation data", but the paper never says this, and the authors helpfully re-state that the data is real-world. The negative review further questions whether the paper's claims will generalize outside the provided dataset, and the authors defuse this concern in the rebuttal by additionally testing on some UAV and 2D data. In sum, the AC congratulates the authors on their good rebuttal, and recommends acceptance.